# Glycopeptide database search and de novo sequencing with PEAKS GlycanFinder enable highly sensitive glycoproteomics

Weiping Sun [1,6], Qianqiu Zhang[2,6], Xiyue Zhang[1,6], Ngoc Hieu Tran [1,2,6], M. Ziaur Rahman[1], Zheng Chen[1], Chao Peng[3,4], Jun Ma[1], Ming Li [2,5] ✉, Lei Xin [1] ✉ & Baozhen Shan[1] ✉

Here we present GlycanFinder, a database search and de novo sequencing tool for the analysis of intact glycopeptides from mass spectrometry data. GlycanFinder integrates peptide-based and glycan-based search strategies to address the challenge of complex fragmentation of glycopeptides. A deep learning model is designed to capture glycan tree structures and their fragment ions for de novo sequencing of glycans that do not exist in the database. We performed extensive analyses to validate the false discovery rates (FDRs) at both peptide and glycan levels and to evaluate GlycanFinder based on comprehensive benchmarks from previous community-based studies. Our results show that GlycanFinder achieved comparable performance to other leading glycoproteomics softwares in terms of both FDR control and the number of identifications. Moreover, GlycanFinder was also able to identify glycopeptides not found in existing databases. Finally, we conducted a mass spectrometry experiment for antibody N-linked glycosylation profiling that could distinguish isomeric peptides and glycans in four immunoglobulin G subclasses, which had been a challenging problem to previous studies.

Protein glycosylation is one of the most prevalent and sophisticated post-translational modifications (PTMs) that have a large impact on many biological processes[1–3]. Glycosylation refers to the process of attaching carbohydrates of diverse structures, called glycans, to different sites of the proteins. The attached glycans may promote the folding and stability of the proteins and play an important role in regulating their functions[4]. Dysregulation of glycosylation, on the other hand, may be associated with various diseases, including congenital disorders, cancers, autoimmune diseases, etc. Thus, comprehensive glycosylation profiling is essential for the development of diagnostic tests and potential treatments[1,5,6], e.g. glycoengineering for therapeutic antibodies[7,8] or spike protein glycosylation for SARS-CoV-2 vaccine development[9,10].

The aim of glycoproteomics[2] is to provide a detailed characterization of glycosylation events in biological samples at a system-wide level, e.g. cell, tissue, or organism. Recent advances of liquid chromatography with tandem mass spectrometry (LC−MS/MS) in glycoproteomics have enabled site-specific profiling of intact glycopeptides, including the protein carriers, the modification sites, the glycan structures, and their quantifications. However, accurate identification of intact glycopeptides from tandem mass spectra still remains challenging and lacks behind the performance of peptide identification[3]. The presence of both glycans and peptides make the fragmentation process, the mass spectra, and their annotations much more complicated. Different types of fragment ions, including peptide b/y, c/z or glycan B/Y ions, can be observed in a mass spectrum,

[1]Bioinformatics Solutions Inc., Waterloo, Ontario, Canada. [2]David R. Cheriton School of Computer Science, University of Waterloo, Waterloo, Ontario, Canada. [3]BaizhenBio Inc., Wuhan, China. [4]Wuhan BioBank, Wuhan, China. [5]Henan Academy of Sciences, Zhengzhou, Henan, China. [6]These authors contributed equally: Weiping Sun, Qianqiu Zhang, Xiyue Zhang, Ngoc Hieu Tran. ✉ e-mail: mli@uwaterloo.ca; lxin@bioinfor.com; bshan@bioinfor.com

depending on the fragmentation strategy, such as resonance-activation collision-induced dissociation (CID), higher-energy collisional dissociation (HCD), electron-transfer dissociation (ETD), or their hybrid combinations. Ambiguous annotations of fragment ions may happen even with very high resolutions of modern MS instruments. Last but not least, it is essential to address the heterogeneity problems that there may be multiple glycosylation sites per peptide and multiple structures with the same glycan composition or mass.

In this paper, we propose GlycanFinder, a glycoproteomics software package comprising database search and de novo sequencing solutions for the identification of intact N-linked and O-linked glycopeptides. We apply both peptide-based and glycan-based search strategies in an independent fashion to pick up all possible candidates from the protein and glycan databases, as long as either their peptide or glycan ions can be observed in a spectrum. This approach reduces the chance that some candidates may be missed due to poor signals of peptide or glycan fragment ions, as opposed to using one strategy alone[11–14]. To further improve the identification, a deep learning model is designed to perform N-linked glycan de novo sequencing on the spectra that cannot be identified by the database search. The tree structures of glycans certainly complicate the fragmentation process and the de novo search space, but on the other hand, they can also be learned to provide valuable features on top of the fragment ions to assist the de novo sequencing. Here we propose a model that integrates dynamic programming, graph neural network[15], and Transformer neural network[16,17] to learn glycan structures and to reconstruct glycan trees from the mass spectra. While trying to boost the sensitivity, one should pay even more attention to the false discovery rate (FDR), especially for glycopeptide spectra where several types of fragment ions are mixed together and increase the chance of observing random matches. GlycanFinder accurately estimates the FDRs at both peptide and glycan levels by following the target-decoy approach and using suitable decoys for those levels.

We performed experiments to evaluate the performance of GlycanFinder, including FDR validation and benchmarks proposed in a recent community-based evaluation study of the HUPO Human Glycoproteomics Initiative[3]. Our results show that GlycanFinder achieved comparable performance to other leading glycoproteomics softwares in terms of both FDR control and the number of identifications. Moreover, GlycanFinder was also able to identify glycopeptides not found in existing databases. We also demonstrated an N-linked glycosylation profiling of four immunoglobulin G subclasses that could distinguish isomeric peptides and glycans, which had been a challenging problem to previous studies[18–20].

## Results

### GlycanFinder workflow for intact glycopeptide analysis

Figure 1 illustrates the workflow of GlycanFinder for intact glycopeptide analysis. GlycanFinder applies both peptide-based[12] and glycan-based[14] searches in order to increase the pool of candidates and hence its sensitivity. In the peptide-based search, a glycopeptide spectrum is quickly searched against the protein database to identify candidate peptides, and candidate glycans are then selected from the glycan database based on the precursor mass and the respective candidate peptides. If the spectrum remains unidentified after the peptide-based search, it proceeds to the glycan-based search which first searches for candidate glycans and then infers candidate peptides. The combination of peptide-based and glycan-based searches utilize both peptide and glycan fragment ions and hence reduces the chance that some candidates may be missed due to poor fragmentation signals.

Once the candidate glycopeptides are identified, a second-round scoring is performed using a comprehensive set of peptide-backbone ions, glycopeptide Y-ions and B ions to evaluate the glycopeptide-spectrum matches (glycoPSMs) and to estimate their FDRs. For peptide FDR, we use the standard target-decoy approach[21] where the

decoy protein database is generated by randomly shuffling the target protein database. However, for glycan FDR, it is not trivial to generate decoy glycans due to their non-linear structures[2]. Instead, we apply random mass shifts of fragment ions to create decoy spectra, as proposed in Fang et al.[22]. A glycoPSM passes 1% FDR filter only if both of its peptide and glycan FDRs are less than or equal 1%. Supplementary Fig. 1a shows an example of the score distributions of target and decoy glycoPSMs for FDR calculation. It should be noted that while this random-mass-shift approach is commonly used, it may underestimate the rate of false matches to glycans with shared fragment ions. Better approaches to generate decoys for glycan FDR estimation will be an interesting topic for future research.

A common problem in glycopeptide analysis is that there may be multiple glycosylation sites in a peptide sequence or multiple glycans with the same composition, which increase the ambiguity of glycoPSM assignments. If the peptide of a glycoPSM has multiple possible glycosylation sites, their localization scores are calculated in GlycanFinder using their respective internal fragment ions. The top-scoring site is then selected and a site-specific localization score, named A-score, is calculated as the score difference between the best and second-best sites. Similarly, when multiple glycans with the same composition match a spectrum, their structure scores are calculated using their respective glycopeptide Y-ions. The top-scoring glycan is then selected and an S-score is calculated as the score difference between the best and second-best glycans. The A-score and S-score of a glycoPSM reflect the confidence of its glycosylation site and glycan structure assignments, as larger differences between the best and second-best scores imply stronger supporting evidence for the assignments with the best scores (Methods).

### De novo sequencing of intact N-linked glycopeptides with deep learning

For those spectra in which the peptides are confidently identified yet no matched glycan can be found from the glycan database, we perform de novo sequencing to identify potentially new N-linked glycans (Fig. 1c). Several glycan de novo sequencing tools have been proposed in previous studies[14,22,23]. For instance, pGlyco3[14] tried to find a modification on Hex, e.g. Hex with an ammonia adduct, and to identify N-linked and O-mannose glycopeptides with that modification. Glyco-Decipher[22] proposed to modify the glycans in an existing database through a process named "monosaccharide stepping" to reveal a new composition or modification that could explain the difference between an observed mass and the mass of an existing glycan. StrucGP[23] reconstructed the structure of a glycan from three predefined modules, including four core structures, three glycan subtypes and 17 branch structures. Unlike previous methods, our glycan de novo sequencing follows a data-driven approach and applies a machine learning model to learn glycan structures from the training data. The trained model is then used to reconstruct glycan trees from scratch, without imposing predefined rules, structures, or types of modifications.

In particular, given a spectrum and a glycan mass (deduced from precursor mass and peptide mass), we use a dynamic programming algorithm to compute the glycan composition. The glycan tree is then constructed from root to leaves, starting from the peptide (root) and iteratively adding monosaccharides (leaves) to the tree. At each iteration, a deep learning model is used to predict the next monosaccharides based on the spectrum and the partial tree obtained from the previous iteration. Each of five monosaccharides (Hex, HexNAc, Fuc, NeuAc, NeuGc) or their combinations are added to the partial tree to create a pool of candidate trees, and then two neural networks are applied to each candidate tree. The first neural network, a Transformer neural network for graphs, i.e. Graphormer[15–17], captures the structures of the candidate trees. The second neural network captures the matched glycopeptide Y and B ions between the candidate trees and the

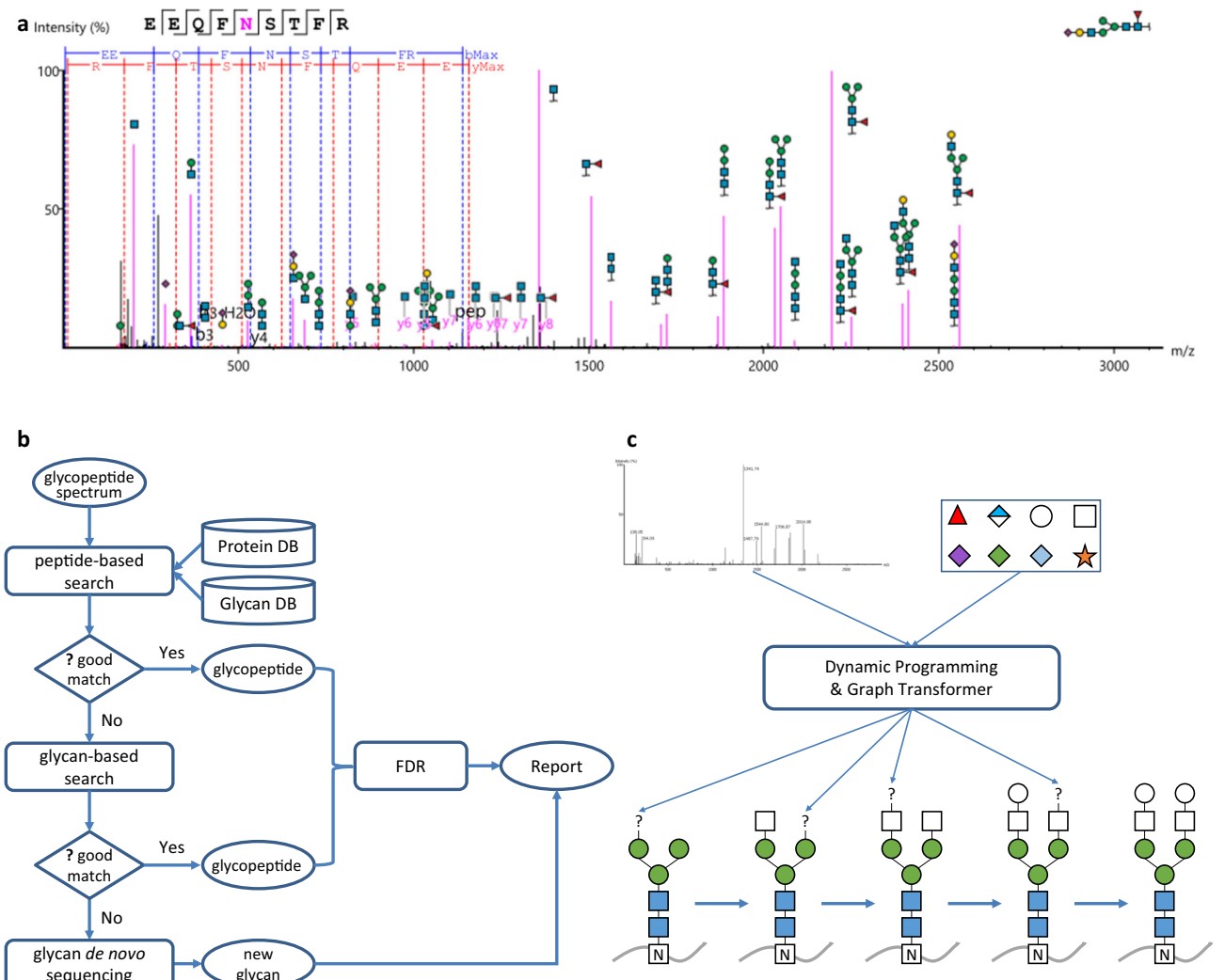

**Fig. 1 | Workflow of GlycanFinder. a** Example of an N-linked glycopeptide-spectrum match. The glycosylation site is indicated by letter N in the purple color of the peptide sequence. The glycan is shown at the top-right corner. Peptide backbone *b/y* ions are highlighted in blue and red, respectively. Glycopeptide *B/Y* ions are highlighted in purple. **b** Workflow of glycopeptide database search and de novo sequencing. A glycopeptide spectrum can be processed in three stages: peptide-based search, glycan-based search, and de novo sequencing. If the spectrum cannot be identified in one stage, it proceeds to the next stage. Once a candidate glycopeptide is identified, the false discovery rates (FDRs) at both peptide and glycan levels are calculated. **c** N-linked glycan de novo sequencing. An N-linked glycan tree is constructed from the N-linked core by iteratively adding mono-saccharides to the tree. At each iteration, a dynamic programming algorithm coupled with a Graph Transformer neural network are used to predict the next monosaccharides based on the input spectrum and the partial tree obtained from the previous iteration. The N-linked core includes two HexNAc (blue squares) and three Hex (green circles). The symbols at the top-right corner indicate different types of monosaccharides that can be added to the glycan tree. Source data are provided in the Source Data file.

spectrum. Their outputs are combined to score the candidate trees and a number of top-scoring trees are selected and fed to the next iteration. The iterations continue until all monosaccharides of the glycan composition computed by the dynamic programming have been used. The key idea of our model is that the structures of the candidate trees, captured by the Transformer graph neural network, can offer additional features on top of the fragment ions to predict the next monosaccharides, as we have demonstrated successfully for peptide de novo sequencing[24–26].

More details of the database search and de novo sequencing of GlycanFinder can be found in the Methods section.

### Evaluation of peptide and glycan FDR estimation
The first and probably most important test to evaluate proteomics search engines is how accurate their FDR estimation is. This is even more critical for glycoproteomics as both glycan and peptide FDRs need to be accounted for. Here we applied an FDR validation approach introduced by Liu et al.[27] and Zeng et al.[14]. N-linked glycopeptide ana-lysis was performed on a dataset of fission yeast glycoproteome samples[27]. The protein database was created by combining fission yeast and mouse proteomes from Swiss-Prot (*Schizosaccharomyces pombe* and *Mus musculus*, respectively). The glycan database contains 1670 glycan compositions that have been used in previous studies[14,28] for evaluating this dataset. The peptide FDR was estimated as the proportion of identified glycoPSMs with mouse peptides. This two-species entrapment approach has also been used in other studies for FDR estimation[29,30]. The glycan FDR was estimated as the proportion of identified glycoPSMs with glycans containing NeuAc, NeuGc, or Fucose, as they are not expected in fission yeast samples[14,27,28].

The results of GlycanFinder were compared against three other tools, including pGlyco3[14] (build 20210615), MetaMorpheus[13] (ver-sion 0.0.320), and MSFragger[12,28] (version 19.0). All tools were run using the same databases and parameters, including 1% peptide and glycan FDRs, Cys(Carbamidomethylation) as fixed modification,

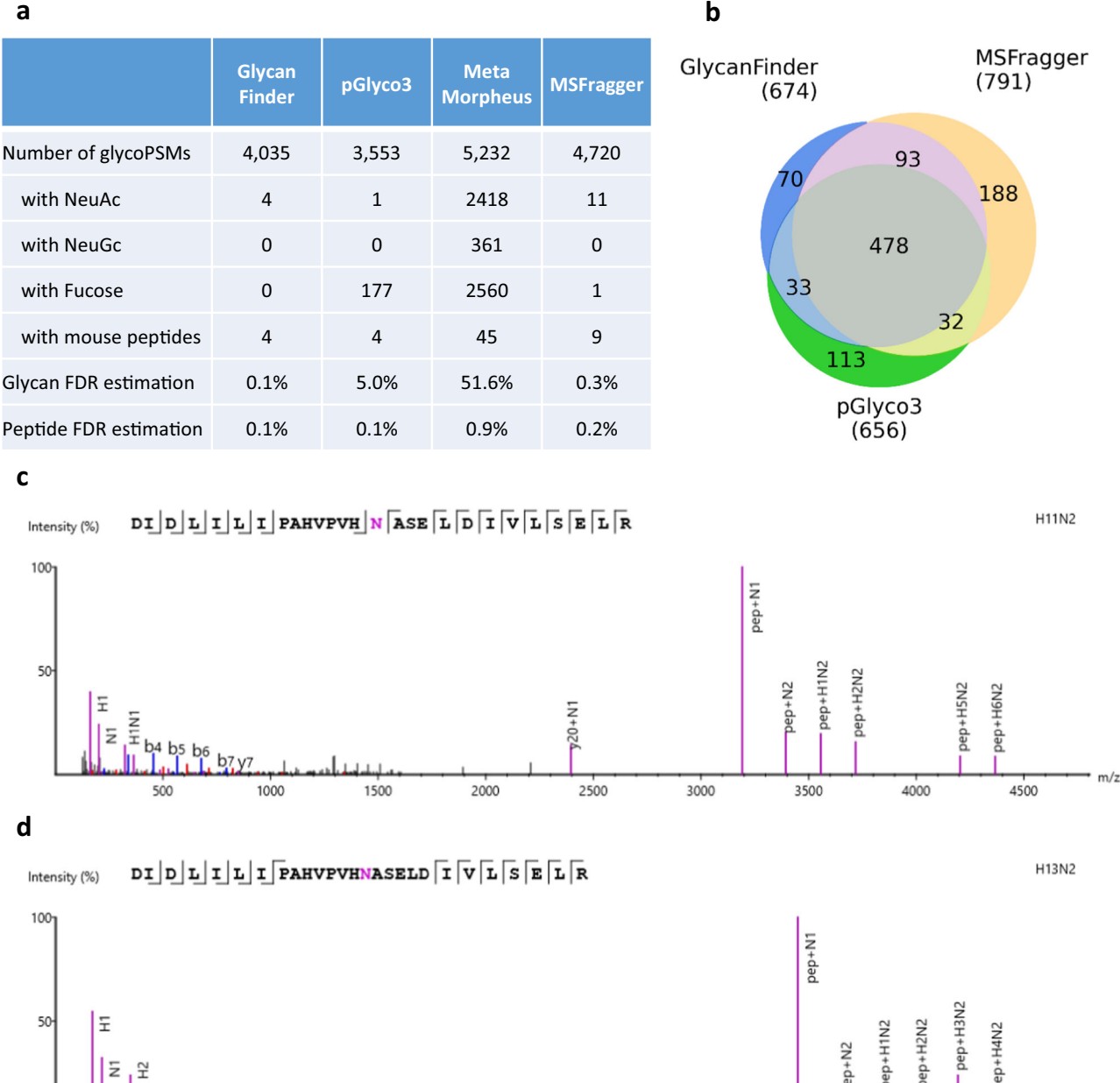

**a**

|  | Glycan Finder | pGlyco3 | Meta Morpheus | MSFragger |
|---|---|---|---|---|
| Number of glycoPSMs | 4,035 | 3,553 | 5,232 | 4,720 |
| with NeuAc | 4 | 1 | 2418 | 11 |
| with NeuGc | 0 | 0 | 361 | 0 |
| with Fucose | 0 | 177 | 2560 | 1 |
| with mouse peptides | 4 | 4 | 45 | 9 |
| Glycan FDR estimation | 0.1% | 5.0% | 51.6% | 0.3% |
| Peptide FDR estimation | 0.1% | 0.1% | 0.9% | 0.2% |

**Fig. 2 | Performance evaluation of glycopeptide database search engines on the fission yeast dataset from Liu et al.[27]. a** Number of identified glycopeptide-spectrum matches (glycoPSMs) and the estimated false discovery rate (FDRs). The glycan FDR was estimated as the proportion of glycoPSMs with NeuAc, NeuGc, or Fucose, as those monosaccharides were not expected in fission yeast. The peptide FDR was estimated as the proportion of glycoPSMs with mouse peptides. **b** Unique glycopeptides identified by GlycanFinder, MSFragger, and pGlyco3. **c, d** Supporting glycoPSMs of the two high-mannose glycans, (HexNAc)2(Hex)11 and (HexNAc)2(Hex)13, which were reported only by GlycanFinder at the glycosylation site N234 of the protein O13781|YEO3_SCHPO. Peptide-backbone *b/y* ions are highlighted in blue and red, respectively. Glycopeptide *B/Y* ions are highlighted in purple. Source data are provided in the Source Data file. (H: Hex; N: HexNAc).

Met(Oxidation) as variable modification. The search results are provided in the Supplementary Data 1. As shown in Fig. 2a, GlycanFinder identified 4035 glycoPSMs, which was 13.6% more than pGlyco3 (3553) and 17.0% less than MSFragger (4720). MetaMorpheus identified the most number of glycoPSMs (5232), but it had a highly elevated glycan FDR of 51.6%. The estimated glycan FDRs of GlycanFinder and MSFragger were 0.1% and 0.3%, respectively, while that of pGlyco3 was 5.0% mainly due to glycans containing Fucose. The peptide FDRs of all tools were well-controlled below 1%. The Venn diagram in Fig. 2b shows that GlycanFinder identified 674

glycopeptides, which was 2.7% higher than pGlyco3 (656) and 17.4% less than MSFragger (791). Since we noticed that our estimated FDRs were too tight, e.g. 0.1% for both glycan and peptide levels, we also attempted to relax the score thresholds so that our FDRs were comparable to those of MSFragger, i.e. 0.3% and 0.2% at glycan and peptide levels, respectively. With those comparable FDRs, GlycanFinder would identify 4518 glycoPSMs, about 4.5% less than MSFragger. In addition, it is worth noting that MSFragger is based on glycan compositions, whereas GlycanFinder and pGlyco3 report glycan structures, which provide more comprehensive information

about the glycans as one composition may correspond to multiple structures.

We also investigated 70 glycopeptides that were identified only by GlycanFinder. Figs. 2c, d show the glycoPSMs identified at the glycosylation site N234 on the protein PO13781|YEO3_SCHPO, which had been reported by Zielinska et al.[31] using a deglycosylation approach (i.e. only the glycosylation site reported but not the specific glycans). GlycanFinder identified three glycans at this site, including (HexNAc)2(Hex)11 and (HexNAc)2(Hex)13 with high-mannose structures (HexNAc)2(Hex)n that are commonly observed in the fission yeast species. We also observed that, for the spectrum scan 56,805 of sample 2 in this dataset, GlycanFinder identified the high-mannose glycan (HexNAc)2(Hex)11 with better supporting glycopeptide *B/Y* ions than pGlyco3's identification on this same spectrum (Supplementary Fig. 1, Supplementary Data 1). MSFragger did not report any identification at this glycosylation site.

We also compared the running times of the four tools on this fission yeast dataset. The analysis was performed on a Dell Precision 7920 Tower with Intel Xeon and 128 G RAM. All four tools support multiprocessors to accelerate the search speed and 36 processors were used in this analysis. All tools read in the same raw files and searched the same protein database and glycan database. Their running times are reported in Supplementary Data 1. The results show that MSFragger was the fastest tool and took 30 min to finish the analysis. GlycanFinder and pGlyco3 completed the analysis in 78 and 148 min, respectively, while MetaMorpheus took much longer time.

## Evaluation of glycan de novo sequencing

We evaluated our deep learning model for glycan de novo sequencing on a dataset of five mouse tissues (brain, heart, kidney, liver, lung) published previously by Liu et al.[27]. N-linked database search was first run on this dataset and 139,208 glycoPSMs identified at 1% FDR were subsequently used for training and testing. Fivefold cross-validation was performed, where the glycoPSMs of four tissues were used for training and the glycoPSMs of the remaining tissue were used for testing. For instance, when the lung data was used for testing, the other four (brain, kidney, heart, liver) were used for training. Similarly, when the brain data was used for testing, the other four (lung, kidney, heart, liver) were used for training. Furthermore, we excluded all glycans of the testing set from the training set to make sure that the training and testing sets did not share any common glycans. Due to this exclusion, the testing was performed on fractions 1 of the five tissues, as using all fractions for testing would substantially reduce the training set. More details of the training process can be found in the Methods section. In addition, we also compared the performance of our glycan de novo sequencing model to StrucGP[23] on this dataset.

For each glycoPSM, the de novo glycan was compared to the target glycan (identified from the database search) based on three levels: composition, fragment ions, and structure. The first level calculated whether the de novo and target glycans had the same composition. The second level calculated the number of matched glycan fragment ions between the de novo and target glycans. The third level calculated whether the de novo and target glycan trees were exactly matched. The testing de novo accuracies of GlycanFinder and StrucGP on five mouse tissues are summarized in Fig. 3a, the detailed results are provided in Supplementary Data 2. GlycanFinder achieved average accuracies of 32%, 83%, and 89% at the three levels structure, fragment ions, and composition, respectively, whereas the accuracies of StrucGP were 23%, 84%, and 85%. While both tools showed comparable accuracies of fragment ions and composition, the structure accuracy of GlycanFinder was substantially higher than that of StrucGP in average and also across all five tissues. These results demonstrate the advantage of GlycanFinder's deep learning model to learn and predict the tree structures of de novo glycans. Fig. 3b and Supplementary Fig. 2 further show two examples of different de novo glycans predicted by GlycanFinder and StrucGP on the same spectra. The glycans predicted by GlycanFinder matched with the respective database search results. In the first example (mouse brain, fraction 1, scan 39,012), StrucGP predicted a glycan with three Fucoses, which did not likely represent a correct structure. In the second example (mouse lung, fraction 1, scan 56,136), the glycan predicted by StrucGP did not contain NeuAc but its signals were presented in the spectrum (Supplementary Fig. 2). Fig. 3c shows two examples of de novo glycans that were only discovered by GlycanFinder and were not found in the database.

We also attempted to provide an FDR estimation for glycoPSMs identified from the glycan de novo sequencing. A proper way to do this is to add the de novo glycans to the original glycan database and then repeat the glycopeptide database search with the new glycan database and FDR control. In particular, we combined 922 de novo glycans identified by GlycanFinder from the testing data and 7884 glycans in the original database, resulting in a new database of 8806 glycans. A second-round database search was then performed using the new database and 1% FDR. We found 1948 additional glycoPSMs corresponding to 389 de novo glycans that passed 1% FDR of the second-round database search, i.e. about 6.8% more glycoPSMs and 4.9% more glycans than the original database search results. Those extra de novo glycoPSMs are provided in the Supplementary Data 2.

A recent study by Rui et al.[26] shows that, for de novo peptide sequencing, the amino acid accuracy could reach 70–80% and the peptide accuracy could reach 40–60%. However, it should be noted that the tree structures of glycans are much more complex than the linear structure of peptides. During the de novo sequencing of a glycan, there are many ways that a monosaccharide can be added to a partial tree, creating multiple branches, whereas for a peptide, amino acids are simply added to the sequence one after another. Here we demonstrated that the tree structures of glycans could be learned by a graph neural network, whereas previously a long short-term memory neural network was used to learn the sequence patterns of peptides. Our results thus reaffirmed that deep learning is essential to learn intrinsic features of peptides and glycans. It will be exciting to see future works that can apply deep learning to predict physicochemical properties of glycopeptides, such as fragment ions, retention times, collisional cross sections, etc.[32–34].

## Antibody N-linked glycosylation profiling

Profiling of immunoglobulin G (IgG) glycosylation has been a challenging problem due to multiple factors[18–20]. The four subclasses of IgG, IgG1–4, have different levels of concentration in the serum yet they share highly similar amino acid sequences, which together significantly affect the separation and detection sensitivity of LC–MS/MS. Here we reported an N-linked glycosylation profiling that could distinguish the four subclasses IgG1–4, without using any special strategies for sample preparation, labeling, or fragmentation[20]. An LC–MS/MS experiment was performed to study the N-linked glycosylation of IgG1–4 proteins. The IgG sample was analyzed on two different MS instruments, Orbitrap (Thermo Fisher Scientific) and timsTOF (Bruker Daltonics), to evaluate the identification and label-free quantification. The LC–MS/MS data was subsequently imported into GlycanFinder, pGlyco3, and MSFragger for N-linked glycopeptide analysis. More details of the sample preparation and the LC–MS/MS analysis can be found in the Methods section.

Figure 4 shows a summary of GlycanFinder results on the IgG Orbitrap dataset. The glycoPSMs in Fig. 4a and Supplementary Fig. 3 show an example where GlycanFinder was able to differentiate the four IgG glycopeptides EEQ[Y/F]N(HexNAc4Hex3Fuc1)ST[Y/F]R, especially the two isomers IgG3 and IgG4. The glycopeptide *B/Y* ions with higher masses and intensities allowed to identify the glycan structure (Supplementary Fig. 3a), whereas the peptide-backbone *b/y* ions with lower masses and intensities helped to identify the

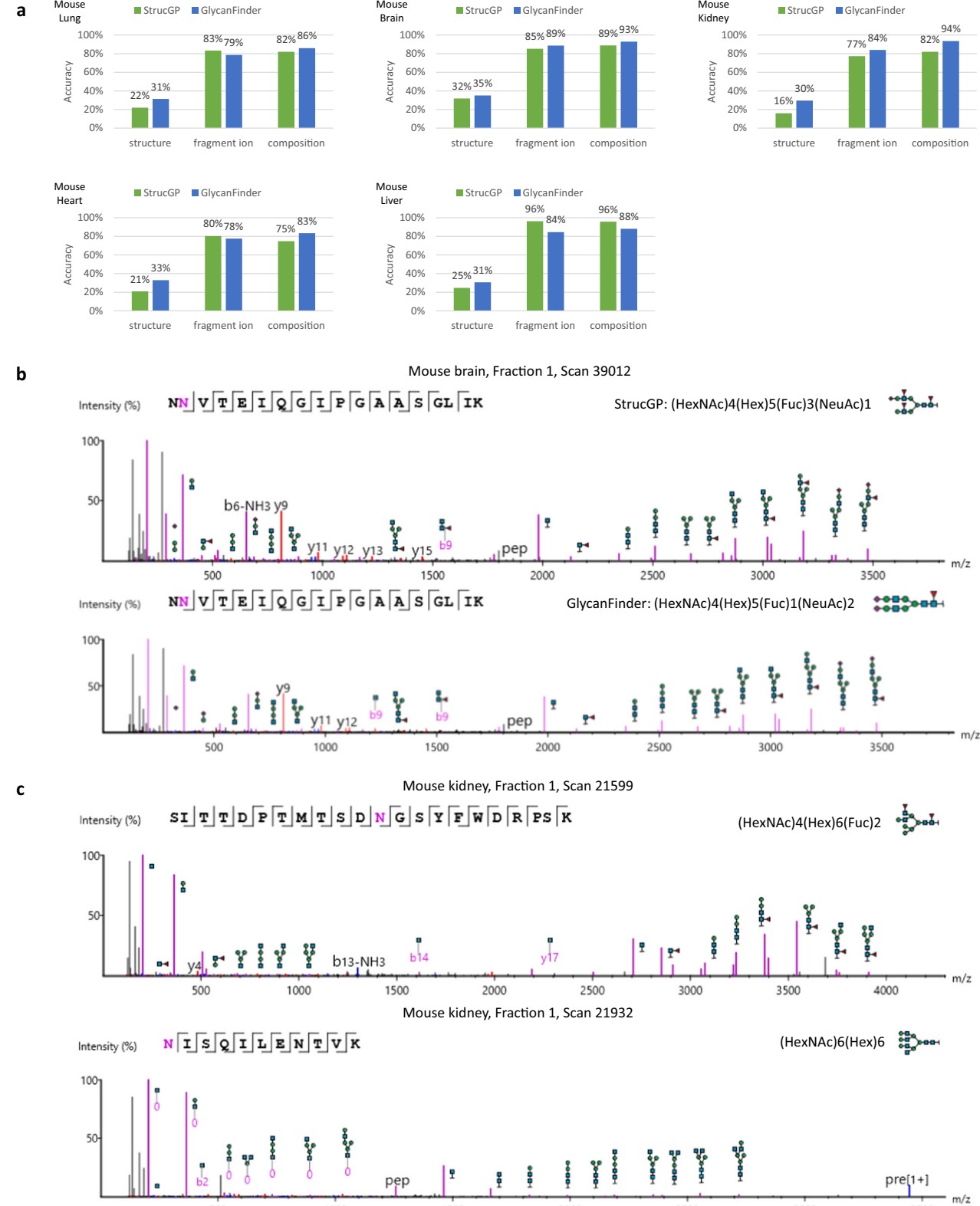

peptide sequences (Fig. 4a). The key ions *b3/b4, y5/y6, b7/b8*, and *y1/y2* corresponding to Tyr (Y) and Phe(F) amino acids at the 4th and 8th positions allowed to distinguish the four glycopeptides of IgG1–4 (Fig. 4a and Supplementary Fig. 3b). Such analysis previously would require a special chemical labeling strategy plus ETD coupled with HCD fragmentations to achieve[20]. It should be noted that, for the Caucasian population, IgG2 and IgG3 share the same tryptic

N-linked glycopeptides, making it impossible to distinguish them[35,36].

In total, GlycanFinder identified 178 unique glycopeptides on the four IgG1–4 subclasses. Fig. 4b, c further show the quantification results of N-linked glycans on the four IgG1–4 subclasses and on two biological replicates. In particular, Fig. 4b shows the N-linked glycan quantification profile of IgG1, where the top five most abundant

**Fig. 3 | Evaluation of the glycan de novo sequencing model of GlycanFinder on the dataset of five mouse tissues from Liu et al.[27]. a** Comparison of the glycan de novo sequencing accuracies of StrucGP and GlycanFinder. The composition accuracy was calculated as the proportion of glycoPSMs in which the de novo and target glycans had the same composition. The fragment ion accuracy was calculated as the proportion of matched glycan fragment ions between the de novo and target glycans in all glycoPSMs. The structure accuracy was calculated as the proportion of glycoPSMs in which the de novo and target glycan structures were exactly

matched. **b** GlycoPSMs of different glycans predicted by StrucGP and GlycanFinder on the same spectrum. The glycan predicted by GlycanFinder matched with the respective database search result, whereas StrucGP predicted a glycan with three Fucoses, which may not likely represent a correct structure. **c** GlycoPSMs of the de novo glycans that were discovered by GlycanFinder and not found in the database. Peptide backbone *b/y* ions are highlighted in blue and red, respectively. Glycopeptide *B/Y* ions are highlighted in purple. Source data are provided in the Source Data file.

glycans are (HexNAc)4(Hex)4(Fuc)1, (HexNAc)4(Hex)5(Fuc)1, (HexNAc)4(Hex)3(Fuc)1, (HexNAc)4(Hex)5(Fuc)1(NeuAc)1, and (HexNAc)5(Hex)4(Fuc)1. Fig. 4c shows how the abundance, defined as the normalized LC–MS precursor feature area, of those glycans change across the four IgG1–4 subclasses. While IgG1 and IgG2 show similar N-linked glycan quantification profiles, IgG3 and IgG4 show substantial different profiles from those of IgG1 and IgG2. We also validated that the glycan quantification profiles were consistent across two biological replicates.

It should be noted that the observed glycosylation profiles depend on the source of the sample under investigation[2]. We also performed the same analysis on the IgG timsTOF dataset and found that, while the timsTOF results contained less number of unique glycopeptides than the Orbitrap results (121 and 178, respectively), the top five most abundant glycans and their relative quantifications were consistent on both datasets (Supplementary Fig. 4, Supplementary Data 3, Supplementary Data 4). Isomeric glycopeptides of IgG1–4 could also be differentiated in the timsTOF results (Supplementary Fig. 4). Furthermore, we compared the results of GlycanFinder to those of pGlyco3 and MSFragger on the IgG Orbitrap dataset (Supplementary Fig. 5). GlycanFinder, pGlyco3, and MSFragger identified 178, 254, and 88 unique glycopeptides, respectively. All glycopeptides of MSFragger were part of GlycanFinder or pGlyco3 identifications. In terms of quantification, pGlyco3 reported the same top five most abundant N-linked glycans and similar quantification profile to that of GlycanFinder (Supplementary Fig. 5b). MSFragger results were slightly different, with less amount of (HexNAc)4(Hex)3(Fuc)1 and more of (HexNAc)4(Hex)5(Fuc)1(NeuAc)1. Overall, we observed consistent quantification results from the two different instruments and the three search engines. The IgG glycosylation profiling and label-free quantification together shall enable comparative analysis across multiple samples to identify biomarkers for diagnostic tests and potential treatments of immune diseases associated with dysregulation of glycosylation[1,2,5,6].

### Comprehensive performance evaluation based on a community study from HUPO Human Glycoproteomics Initiative

Kawahara et al.[3] recently described a community-based effort of the HUPO Human Glycoproteomics Initiative (HGI) to evaluate the performance of 11 glycoproteomics softwares from 9 developer teams and 13 user teams for intact glycopeptide analysis. Their study provided standard glycoproteomics datasets from human serum and comprehensive criteria to test the performance of those search engines. Here we also evaluated GlycanFinder on the same benchmarks and compared to the results reported in Kawahara et al.[3], including IQ-GPA v2.5[37], Protein Prospector v5.20.23[38], glyXtool^MS v0.1.4[39], Byonic v2.16.16[11], Sugar Qb[40], Glycopeptide Search v2.0alpha[41], GlycopeptideGraphMS v1.0[42], GlycoPAT v2.0[43], and GPQuest v2.0[44]. We also compared our results to those of the best user teams reported in Kawahara et al.[3] (team 15 using Byonic for N-linked and team 13 using Byonic for O-linked).

The evaluation was performed on the dataset HCD-EThCD-CID-MS/MS (file B from Kawahara et al.[3]). N-linked and O-linked glycopeptide analyses were performed using the same protein and glycan databases and the search parameters as in the HGI study (the details are provided in the Methods section). The N-linked and O-linked

glycoPSMs identified by GlycanFinder are provided in the Supplementary Data 5. The search results of the other tools reported in Kawahara et al.[3] were obtained from that study's supplementary materials; this could be a disadvantage to those tools as that study was published about a year before ours. The search results were further used to calculate eleven different criteria N1–N6 and O1–O5 proposed in Kawahara et al.[3] to evaluate the tools' performance. More details of the evaluation criteria can be found in the Methods section. In addition to the normalized scores of those criteria, we also reported direct measurement results according to those criteria (Supplementary Data 5).

The N-linked evaluation results in Fig. 5 show that, overall, GlycanFinder achieved a slightly better performance than the best result (user team 15) reported in Kawahara et al.[3] (0.789 versus 0.777), and outperformed the other nine softwares. Moreover, GlycanFinder consistently scored high across five criteria N1–N3, N5, N6 (0.833–0.952), indicating its high level of accuracy in identifying the expected and consensus N-linked glycopeptides and glycoproteins while properly controlling the false discovery rate. However, GlycanFinder did not perform well on the N4 test in terms of the number of identified N-linked glycopeptides. We further checked the consensus between the results of GlycanFinder and the other tools. Fig. 5b shows that 83% of N-linked glycan compositions and 78% of N-linked glycoproteins of GlycanFinder were also reported by at least three other tools. Fig. 5c also shows a consistent classification of the N-linked glycans reported by GlycanFinder to those reported by other high-scoring tools such as user team 15, Protein Prospector, or Byonic.

### Evaluation of O-linked glycopeptide analysis

GlycanFinder also performs peptide-based and glycan-based searches of O-linked glycopeptides in a similar fashion to that of N-linked glycopeptides. However, unlike N-linked glycopeptides which usually have a N-X-S/T/C (X≠P) motif, O-linked glycans are attached to proteins via the hydroxyl groups of serine (S) or threonine (T) residues. Thus, there are often more than one O-glycosylation sites in a peptide sequence. GlycanFinder allows at most two O-linked glycans per peptide and uses internal fragment ions to determine the best glycosylation sites and to calculate the site-specific localization score (A-score) for its glycosylation site assignments (Supplementary Fig. 6). More details about O-linked glycopeptide analysis and the A-score can be found in the Methods section. The benchmark results for O-linked glycopeptide analysis based on the HGI study are shown in Supplementary Fig. 7. GlycanFinder achieved an overall score of 0.730 and outperformed the other nine softwares as well as the best result (0.701, user team 13) reported in Kawahara et al.[3]. GlycanFinder performed consistently well across four criteria O2–O5 (0.780–1.000) while having a low score for criterion O1 (O-glycan composition). All other tools except IQ-GPA also scored low on this criterion O1, which measures the Pearson correlation between the expected and the observed O-glycan distribution in human serum. The same results for O1 had also been observed in Kawahara et al.[3], where using a narrow search space and permitting only few missed peptide cleavages were suggested to improve O1 score. Overall, the benchmark results based on the HGI study demonstrate that GlycanFinder represents a high-performance informatics solution for both N-glycoproteomics and O-glycoproteomics.

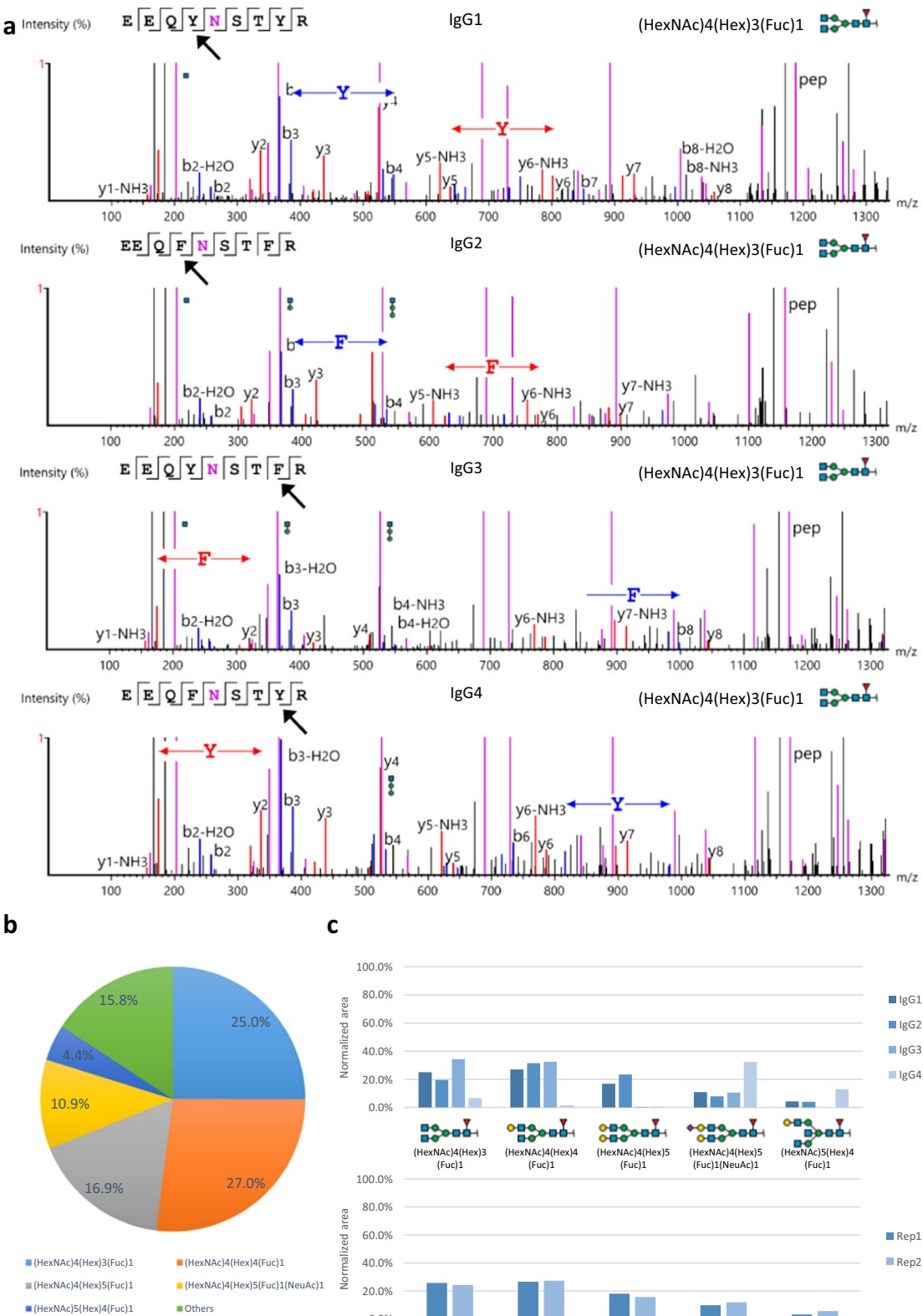

**Fig. 4 | N-linked glycosylation analysis of four immunoglobulin G subclasses IgG1-4.** The glycosylation sites are N180 (IgG1), N176 (IgG2), N227 (IgG3), and N177 (IgG4). **a** Example glycoPSMs of the four isomeric glycopeptides EEQ[Y/F]N(Hex-NAc4Hex3Fuc1)ST[Y/F]R on IgG1-4. The spectra are zoomed into the key peptide-backbone ions *b3/b4, y5/y6, b7/b8*, and *y1/y2* that distinguish Tyr (Y) and Phe (F) amino acids at the 4th and 8th positions. Peptide backbone *b/y* ions are highlighted in blue and red, respectively. Glycopeptide *B/Y* ions are highlighted in purple. The original spectra and more details of the matched fragment ions are provided in Supplementary Fig. 3. **b** N-linked glycan quantification profile at the glycosylation site N180 of IgG1. The percentages indicate the normalized areas of the glycans identified at the site. **c** Normalized areas of the top five most abundant N-linked glycans of IgG1-4 (upper) and of IgG1 on two biological replicates (lower). The quantification of glycans and glycopeptides was based on their MS1 precursors as described in the Methods section. Source data are provided in the Source Data file.

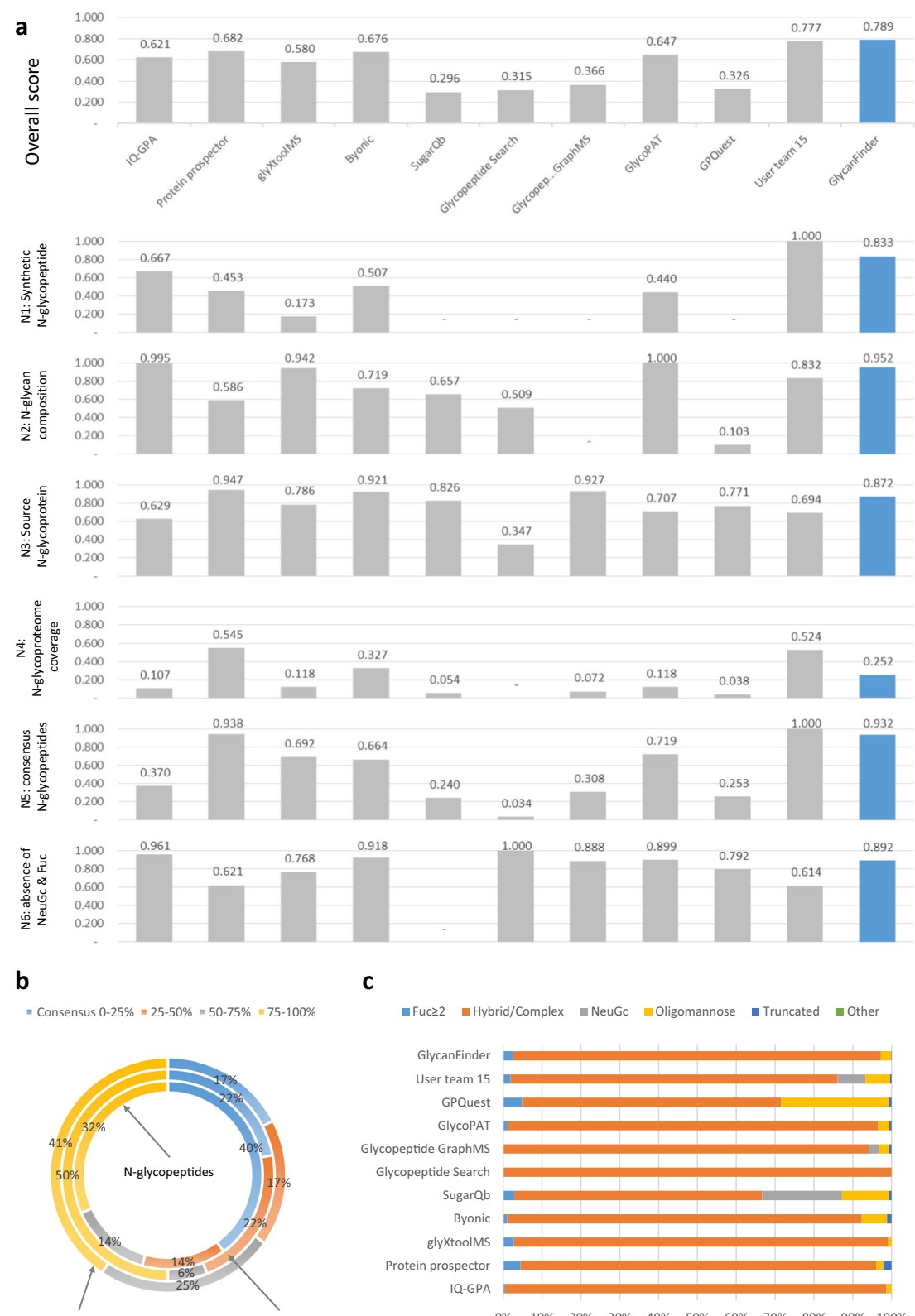

**Fig. 5 | Performance of N-linked glycopeptide database search engines on community-based evaluation benchmarks proposed in Kawahara et al.[3].** **a** Performance of the search engines on six N-linked evaluation criteria and their overall scores. **b** The consensus of identified N-linked glycopeptides, glycoproteins, and glycan compositions between GlycanFinder and the other search engines.

Consensus 0-25% indicates the identifications that are common between GlycanFinder and ≤25% of the other search engines. **c** The classification of N-linked glycans identified by the search engines. Source data are provided in the Source Data file.

## Discussion

In this study, we have demonstrated that GlycanFinder offers a highly sensitive solution for glycoproteomics to boost the identification and to discover new glycopeptides. This is the result of an effective integration of peptide-based and glycan-based search strategies to utilize both peptide and glycan fragment ions to pick up all possible candidates from the databases, mitigating the problem of poor fragmentation signals. The identified glycopeptides are checked for the FDRs at peptide and glycan levels, as well as the accuracies of glycosylation site and glycan structure. The improvements of sensitivity and quality control by GlycanFinder shall enable more glycoproteomics applications in health and diseases, such as therapeutic antibodies or vaccine development where glycans play a key role in the folding and functioning of antibodies. For instance, we have demonstrated an application of GlycanFinder for site-specific glycosylation profiling of four immunoglobulin G subclasses, which can be used to develop biomarkers for diagnostic tests.

In addition to the glycopeptide database search, GlycanFinder also provides a glycan de novo sequencing tool which shall become valuable for glycoproteomics as most current glycan databases are still incomplete. We showed that the tree structures of glycans can be learned by a graph neural network to predict de novo glycans at a comparable accuracy to peptide de novo sequencing. Our results also suggest that deep learning can be applied to predict important features of glycopeptides, such as fragment ions or retention times, which subsequently can be used to improve their identification[2,32–34]. This research direction will see significant applications in glycoproteomics as they can address the difficulties of glycopeptide separation and detection by LC–MS.

Our de novo sequencing is currently only applied to N-linked glycans as we found it much more challenging for O-linked glycans. The O-linked search space is more complicated due to multiple possible occurrences of O-linked glycans per peptide. Furthermore, O-linked glycans appear to have less predictable structures than N-linked and their existing databases are very limited, making it difficult for machine learning models to learn useful features to assist de novo sequencing. Thus, de novo sequencing of O-linked glycans still remains a challenging problem for future research topics.

Our study did not include a comprehensive evaluation of glycopeptide quantitative analysis. In addition to the identification, accurate quantification of intact glycopeptides is essential for differential analysis of site-specific glycosylation. For instance, a recent deep learning-based method, pGlycoQuant[45], has been proposed for intact glycopeptide quantitation and can be used with common search engines such as pGlyco3, MSFragger, or Byonic for quantitative glycoproteomics.

There are some other limitations in our study that could be addressed in future works. For instance, the glycoproteome of fission yeast contains simple high-mannose structures which might be biased and limited for FDR evaluation. The IgG N-linked glycosylation analysis included only one sample which was not enough to account for biological variation. The evaluation benchmarks and results obtained from the HGI study by Kawahara et al.[3] were published about a year before our study. Thus, this could be a disadvantage to those tools and new updates or progress might have been made during that period.

## Methods

### GlycanFinder database search

Potential glycopeptide spectra are separated from non-glycopeptide spectra based on the presence of glycan signature ions and "ion-ladder" patterns. The following oxonium ions are considered: Hex (163 mz), HexNAc (204 mz), Fuc (147 mz), NeuAc (292 mz), NeuGc (308 mz), HexNAc-H2O (186 mz), HexNAc-2H2O (168 mz), NeuAc-H2O (274 mz), and Hex+HexNAc (366 mz). If a spectrum contains 1 signature ion with relative intensity ≥5% or at least 2 signature ions, it is considered as a glycopeptide spectrum. If there is no signature ion, we further look for glycan "ion-ladder" patterns, which are defined as groups of ≥3 continuous glycopeptide Y-ions that are different by 203, 162 or 146 mz. If there is no signature ion nor ion-ladder in a spectrum, it is considered as a native (i.e. non-glycopeptide) spectrum. Glycopeptide spectra go through the glycopeptide search illustrated in Fig. 1b; native spectra go through a standard peptide search.

After the in silico cleavage of the protein database, peptide sequences are filtered based on the presence of potential glycosylation sites. Peptides with the sequons N-X-S/T/C (X≠P) for N-linked and S/T for O-linked glycans are selected for the glycopeptide search with glycopeptide spectra. All peptides are used for the standard peptide search with native spectra.

Peptide-based and glycan-based searches are performed to generate candidate glycopeptides. During the peptide-based search, possible glycan peaks are first removed from a glycopeptide spectrum to avoid the influence on the peptide score calculation. Fragment ions at the lower end of a spectrum are quickly searched against peptide-backbone ions, including $b/y$, $b/y + HexNAc$ for HCD and $c/z$ for ETD data, to identify candidate peptides. We also consider the ions with losses of ammonia ($-NH_3$) or water ($-H_2O$), and charge (2+). Each spectrum is matched with the in silico digested peptides according to a peptide scoring function, with glycans treated as PTMs. The peptide scoring function is similar to the normal PEAKS DB search scoring. It uses a linear discriminative function (LDF) score to measure the quality of a peptide-spectrum match. After a peptide with a glycan mass offset is selected based on the peptide score, glycan candidates can be obtained from the glycan database according to the mass. Glycopeptide Y-ions and B ions are then used to calculate the glycan score. The glycan scoring function is obtained by training by an XGBoost regression model on a variety of published datasets. Tens of features are tuned to evaluate a glycan-spectrum match. Several key features include "ratio of observed glycan Y-ions", "$\log_{10}$ Y-ions intensity", "observed core structure", " ratio of observed glycan B ions" and etc.

If the spectrum cannot be identified by the peptide-based search, it proceeds to the glycan-based search. Here the fragment ions at the higher end of the spectrum are quickly searched against glycopeptide Y-ions to identify candidate glycans. Subsequently, candidate peptides are deduced from the precursor mass, candidate glycans, and the protein database. The peptide mass is further added to glycan fragments when matching glycan Y-ions with the spectrum. Glycopeptide-spectrum matches are then evaluated using the same peptide and glycan scoring functions as in the peptide-based search.

The FDR estimation includes both peptide and glycan FDRs. For peptide FDR, a decoy protein database is generated by randomly shuffling the target protein database. The combined target+decoy protein database is then used for the glycopeptide search. The glycoPSMs are sorted according to the peptide score and the peptide FDR is calculated as

$$\text{peptide FDR} = \frac{\text{number of glycoPSMs with decoy peptides}}{\text{number of glycoPSMs with target peptides}} \quad (1)$$

For glycan FDR, we apply random mass shifts of 3–30 mz to all fragment ions in an MS2 spectrum to create a decoy spectrum. The glycoPSMs are sorted according to the glycan score and the glycan FDR is calculated as:

$$\text{glycan FDR} = \frac{\text{number of glycoPSMs with decoy spectra}}{\text{number of glycoPSMs with target spectra}} \quad (2)$$

The thresholds for peptide and glycan scores are determined from the input peptide and glycan FDR filters. A glycoPSM passes 1% FDR filter only if both of its peptide and glycan FDRs are less than or equal 1%.

When multiple glycan structures with the same composition match a spectrum, they are sorted according to their glycan scores. The top-scoring structure is then selected and the S-score of the glycoPSM is defined as

$$S\text{-score} = \frac{\text{top glycan score} - \text{second glycan score}}{\text{top glycan score}} \quad (3)$$

$$T[m] = \begin{cases} -\inf, & \text{if } T[m - m_{\text{mono}}] < 0 \text{ for all monosaccharides} \\ \text{Intensity}[m] + \max\left(T[m - m_{\text{mono}}]\right) \text{ for all monosaccharides}, & \text{otherwise} \end{cases} \quad (4)$$

**Site-specific localization score (A-score) and O-linked glycopeptide analysis.** If the peptide of a glycoPSM has multiple possible glycosylation sites, a site-specific localization score, named A-score, is calculated by comparing those glycosylation sites and their respective internal fragment ions. In order to distinguish the ambiguity between potential glycosylation sites, fragment ions exclusive to a specific site need to be identified in order to assign a glycan to the residue. In this study, by "internal fragment ions" we refer to those ions that were fragmented more than once, such as a peptide ion with a glycan fragment attached, or a peptide internal ion with an intact glycan attached. In practice, GlycanFinder considers peptide ions with one monosaccharide attached (i.e. the glycan root) or with an intact glycan attached when calculating the glycopeptide score and the A-score. The ion types depend on the fragmentation methods. For example, *b/y* ions + glycan fragments for HCD, *b/y/c/z/z'* ions + glycan fragments for EThCD, and *c/z* ions + glycan fragments for ETD fragmentation.

The A-score in GlycanFinder is calculated as $-10 \times \log_{10} P$, where the *P* value indicates the likelihood that the glycan site is assigned by chance. We adopted this idea from a previous study of phosphorylation site localization by Beausoleil et al.[46], and also from Zhang et al.[47]. More specifically, the probability of the correct glycosylation site is calculated based on the likelihood of identifying site-determining internal fragment ions compared to random chance. An A-score of 42.89 would represent a probability of less than 1 in 15,000 of matching a difference by random chance. If a glycan site can be inferred confidently, for instance, if there can be only one N-glycan site in a peptide, then the A-score would be 1000.

O-linked glycans are attached to proteins via the hydroxyl groups of serine (S) or threonine (T) residues. There are often more than one O-glycosylation sites in a peptide sequence. GlycanFinder allows at most two O-linked glycans per peptide, it is also recommended to provide a glycoprotein database (less than 500 entries) instead of a large, entire proteome database. GlycanFinder considers internal fragment ions to determine the best glycosylation sites and calculates the site-specific localization score (A-score) to reflect the confidence of its glycosylation site assignments as described above (Supplementary Fig. 6).

## GlycanFinder de novo sequencing

**Dynamic programming algorithm to determine the glycan composition.** If a glycopeptide spectrum has its peptide identified but no matched glycan can be found from the glycan database, glycan

de novo sequencing is performed to identify potentially new glycans. The glycan mass is first deduced from the precursor mass and the peptide mass. A dynamic programming algorithm is then used to compute the glycan composition based on the glycan mass and the spectrum. Five common classes of monosaccharides are considered: Hex, HexNAc, Fuc, NeuAc, and NeuGc. All peaks with mass values greater than the peptide mass (i.e. potential glycopeptide Y-ions) in the spectrum are used to compute the composition. The dynamic programming algorithm is designed to avoid recomputation. A table of length *L* is initialized with 0 for the first cell and *−inf* for all others to store the cumulative intensity and the class of monosaccharide residues in the path from peptide mass to precursor mass. Each cell in the table is denoted as *T*[*m*] where m represents mass value.

$$L = \frac{\text{precursor mass} - \text{peptide mass}}{\text{tolerance}} \quad (5)$$

with error tolerance of 0.01 Da.

Finally, we traceback over the table to obtain the best path (i.e. the path has the highest intensity) from precursor mass to the peptide mass and extract the monosaccharides selected at each cell as the glycan composition. The algorithm can run efficiently when using the broadcast behavior to extract peaks within error tolerance in popular Python frameworks such as Numpy.

**Deep learning model to construct the glycan tree.** Once the composition is determined, the glycan tree is constructed from the asparagine (Asn) residue of the peptide, i.e. the root, to leaves by adding various monosaccharides iteratively. At each iteration, a combination of monosaccharides is selected by a deep learning model and attached to a leaf node of the sub-tree obtained from the previous iteration. It should be noted that Fig. 1c only shows a simple example where only one monosaccharide is attached to a leaf node. When a combination consisting of two or more monosaccharides is attached to a leaf node, it will create a branch at that node. The iterative process of adding such combinations will result in a branched topology of the glycan tree where multiple monosaccharides can be linked to any nodes in the tree. The set of all possible combinations of monosaccharides are derived from the training data. The training data in this study consists of twenty-one combinations of five monosaccharides (Hex, HexNAc, Fuc, NeuAc, NeuGc), along with a unique token indicating that no additional monosaccharide could be appended to the leaf node. The monosaccharides and their combinations can be updated when the training data is updated.

The deep learning model consists of two neural networks (Supplementary Fig. 8). The first neural network is applied to encode topologic knowledge of the sub-tree. We use the Graphormer architecture[17] which inherits from the Transformer[16] and is designed for graphs. The glycan structure is represented as a graph with nodes as monosaccharides and edges as linkages, and each of these nodes and edges is assigned a vector embedding. We encode each node in breadth-first search order with derived compositions from dynamic programming without the accumulated monosaccharides. The representations are then passed through two layers and four attention heads to enable the model to learn the relationships between different nodes and edges in the graph. The positional-embedding layer is also

included to allow the model to learn the relative positions of the nodes and edges. The output of this component is a 512-dimensional vector as graph embeddings, which captures the encoded glycan structure.

The second neural network captures the matched glycopeptide Y-ions between the candidate trees and the spectrum. The theoretical spectrum is formed by attaching all candidates to the previously predicted substructure at each time step. The observed spectrum is represented by a set of m/z values and intensity pairs. Similar to PointNovo[26], the pairs are encoded with respect to the mass gap between theoretical m/z and observed m/z. The differences between observed peaks and theoretical peaks are obtained as input features to the point-cloud model named T-Net. The permutation of its monosaccharides is used for each candidate combination as features. For example, the last child monosaccharides of the bisect core structure (Hex2HexNAc1) will have features of Hex, HexNAc, HexHexNAc, Hex-Hex, Hex2HexNAc. At each prediction step, we obtain the theoretical m/z values for candidates and their permutations to compute the m/z difference tensor (D) for the observed peaks. We adopt the activation function from PointNovo to extract features from the spectrum:

$$\sigma(D) = \exp[-|D| \cdot c] \tag{6}$$

The spectrum features are also encoded into 512-dimensional vectors to ensure a fair concatenation of results from the two neural networks.

Finally, representations learned by the two neural networks are concatenated together and fed into a fully-connected layer which adjusts the weights between the two networks. The total number of parameters is 10,939,134. During training, a batch size of 256 and 10 epochs are used with 8-1-1 train-valid-test ratio. The best model is selected based on the least validation error. The model was trained and tested on 25 fractions, with each tissue (mouse brain, lung, liver, kidney, and heart) containing five fractions, in a fivefold cross-validation fashion: the data of four tissues were used for training and the data of the remaining tissue were used for testing. The training loss decreased from 0.220–0.105, while the validation loss also decreased to 0.251. During prediction (de novo sequencing), the two neural networks are used to sort candidates at each time step and the candidate that has the highest rank and contains only the monosaccharides found in the composition (predicted by dynamic programming approach) is selected. The candidate is then attached to the sub-tree and fed back to the neural networks for the next iteration. The iterations continue until the constructed tree includes all monosaccharides in the composition.

**Data and model training.** To train the neural networks, we ran GlycanFinder database search on the dataset of five mouse tissues (brain, heart, kidney, liver, lung) and identified 139,208 glycoPSMs at 1% FDR. However, we found that the number of unique glycan structures was limited, e.g. the mouse brain tissue only contained 1155 unique glycan structures. Learning topologic knowledge on a thousand structures is not sufficient to generalize to other tissues. As a result, we divided the training process into two phases. The first phase is to take advantage of the glycan database and select 5012 unique structures to train the first deep learning model. To guarantee fair comparison, we excluded all glycan structures that exist in the testing set from the training set. For each glycan structure, we iteratively removed all leaves attached to its parent monosaccharides from the first parent node to the root (i.e. in the opposite direction to that of the de novo sequencing). At each iteration, the partition created a sub-tree and the candidate (i.e. a set of monosaccharides) that had been removed. The second phase used the trained parameters from the first deep learning model and trained the second neural network on the spectrum to obtain the next monosaccharides group from 21 candidates. The predicted and the target monosaccharides were used to calculate the training loss, which was

then back-propagated to update the parameter weights of the two neural networks.

## GlycanFinder quantification of glycopeptides

For each identified glycopeptide, the sum of the areas of its MS1 precursor features with different charges is used for its quantification. The label-free approach is applied for glycopeptide quantification among different samples. Given a glycosylation site of a glycoprotein, the glycan profile is calculated based on the normalized areas of the glycan forms at that site. First, the area of each glycan form at each glycosylation site is calculated by summing the areas of all glycopeptides containing that glycan at that site. Then, the normalized area of each glycan form is obtained by dividing its area by the sum of the areas of all glycans forms at that site.

## Evaluation of peptide and glycan FDRs on the fission yeast dataset

The dataset was downloaded from the PRIDE Consortium database. The proteomes of fission yeast and mouse, *S. pombe* and *M. musculus*, respectively, were downloaded from Swiss-Prot (August 19th, 2022) in fasta format. The fasta files were concatenated to create the protein database. The glycan database containing 1670 glycan compositions was obtained from Zeng et al.[14] N-linked glycopeptide database search was performed using GlycanFinder, pGlyco3 (build 20210615), MetaMorpheus (version 0.0.320), and MSFragger (version 19.0) with the following parameters: HCD fragmentation, trypsin cleavage, Cys(Carbamidomethylation) as fixed modification, Met(Oxidation) as variable modification, precursor error tolerance of 10 ppm, fragment error tolerance of 0.05 Da, glycan fragment error tolerance of 20 ppm. The peptide and glycan FDRs were set at 1%. The search results are provided in Supplementary Data 1. glycoPSMs with peptides coming from mouse proteins were counted as false identifications and were used to estimate the peptide FDR. glycoPSMs with NeuAc, NeuGc, or Fucose in their glycan compositions were counted as false identifications and were used to estimate the glycan FDR.

## LC−MS/MS analysis of N-linked glycosylation in the four IgG1−4 subclasses

The IgG sample was purchased from Solarbio Life Sciences (catalog SP001).

**IgG purification and desalting.** IgG was isolated using Protein G column by affinity chromatography, and desalted using Hi-trap Desalting column according to the molecular weight of substances. Serum (20 μL) was diluted in loading buffer (0.1 M, pH 7.4) followed by filtered through a 0.22-μm filtrate with 96-well-plate. Equilibrium buffer and binding buffer are the same as the loading buffer, and 0.1 M formic acid (FA, pH 2.5) was used as the elution buffer. Sample solution was injected into the loop valve and processed following the certain method template of step elution for monoclonal antibody contained in the instructions of AKTA plus. The detailed procedures can be found in the template description of User Manual (AKTA prime plus). Notably, the neutralizing buffer (1 M Tris−HCl, pH 9.0) was suggested to be initially added into the collection tube to maintain the purified IgG fractions stable. According to the principle of molecular sieve, the collected IgG protein solution was further desalted using the desalting column by the application template. IgG desalting was conducted using 50 mM ammonium bicarbonate (pH 8.0) solution due to labile property. Desalting IgG protein solution was automatically collected, and evaporated under a vacuum concentrator at 60°C. Finally, sodium dodecyl sulfate-polyacrylamide gel electrophoresis (SDS-PAGE) was conducted to test the purity of IgG fractions.

**Tryptic digestion and glycopeptides enrichment.** All the IgG samples were subjected to proteolytic cleavage using trypsin. The samples

were dissolved in 150 µL 8 M urea, 0.1 M Tris–HCl (pH = 8.5). Then, 1.5 µL of 0.5 M Tris-(2-carboxyethyl) phosphine (TCEP) was added and incubated at 37 °C and 600 rpm for 30 min. After that, 5.1 µL of 0.3 M iodoacetamide (IAA) solution was added and incubated at 25 °C in the darkness for 30 min. The sample was diluted with 0.1 M Tris (pH 8.5) to 4 times volume of solution. Next, 10 µg of trypsin was added by enzyme to substrate ratio at 1:50 and incubated under 37 °C and 600 rpm for 12 h, and then 5 µg of trypsin was added and incubated under 37 °C and 600 rpm for 4 h. Then, the digestion was stopped by added 1% formic acid and desalted by Sep-Pak 50 mg tC18 column. The peptides mixtures were used for glycopeptide enrichment experiment. Briefly, the column containing 100 mg MCC was pre-washed with 3 mL of ultra-water four times, and equilibrium with 3 mL of 80% acetonitrile (ACN) containing 0.1% trifluoroacetic acid (TFA) four times. The sample was diluted with equilibrium buffer to five times volume of tryptic digestion solution, subsequently loaded onto the SPE column by gravity twice. After washing with 3 mL of 80% ACN/0.1% TFA six times in more than 1 min each time, IgG glycopeptides were collected by 1 mL of ddH2O twice.

**LC-MS/MS experiment.** Subclass-specific IgG glycosylation was analyzed with a nanoElute UHPLC system coupled to a timsTOF Pro2 mass spectrometer equipped with Captive Spray source (Bruker Daltonics). Solvent A and B were 0.1% FA in ultra-water and 0.1% FA in ACN, respectively. Tryptic IgG glycopeptides were dissolved in solvent A and selected for MS analysis. 100 ng of sample was injected and the analytes were separated using a 60 min binary gradient at a flow rate of 300 nL/min. Glycopeptides were separated on a homemade C18 column (75 µm × 20 cm, 1.9um C18 Beads). The LC gradient used was as follows: 0–45 min, 4–22% B; 45–50 min, 22–35% B; 50–55 min, 35–80% B; 55–60 min, 80% B.

For the timsTOF Pro2 settings, the following parameters were adapted, starting from the PASEF method for standard proteomics. Stepping CE was applied for glycopeptides analysis. The values for mobility-dependent collision energy ramping were set to 75 eV at an inversed reduced mobility (1/k0) of 2.0 V s/cm$^2$ and 20 eV at 0.60 V s/cm$^2$. Collision energies were linearly interpolated between these two 1/k0 values and kept constant above or below. For efficient glycopeptide dissociation, TIMS stepping was applied with two collision energies: 35–131.25 eV was utilized following 20–75 eV. 5 PASEF MS/MS scans were triggered per cycle (2.06 s). Target intensity per individual PASEF precursor was set to 100,000. The scan range was set between 0.70 and 1.74 V s/cm$^2$ with a ramp time of 200 ms. Precursor ions in an m/z range between 100 and 4000 with charge states 2–5 were selected for fragmentation. MS was operated in the positive-ion mode, and MS/MS was acquired under the Data dependent acquisition (DDA) mode. Active exclusion was enabled for 0.4 min (mass width 0.015 Th, 1/k0 width 0.015 V s/cm$^2$).

**LC-MS/MS data analysis.** The LC–MS/MS data was imported into GlycanFinder, pGlyco3, and MSFragger for N-linked glycopeptide analysis. The data was searched against an IgG protein database of 9 entries and an IgG N-linked glycan database of 247 entries. The protein and glycan databases and the raw data have been deposited to the PRIDE repository (see "Data availability" section). The following search parameters were used for all three search engines: HCD fragmentation, trypsin digestion, C(Carbamidomethylation) as fixed modification, M(Oxidation) and NQ(Deamidation) as variable modifications, precursor error tolerance of 10 ppm, fragment error tolerance of 0.02 Da, glycan fragment error tolerance of 20 ppm. The peptide and glycan FDR filters were set at 1%.

**Evaluation based on the HGI benchmarks**
The evaluation was performed on the dataset HCD–EThCD–CID–MS/MS (file B from Kawahara et al.[3]). N-linked and O-linked glycopeptide

analyses were performed as follows. The protein database was obtained from the HGI study, which included 20,201 human proteins. The HGI study also provided two default glycan databases that contained 309 mammalian N-glycan compositions and 78 mammalian O-glycan compositions. Since GlycanFinder is a glycan structure based search engine, we collected all the glycan structures from our internal glycan database that had the same compositions as that of the HGI study. As a result, 494 N-glycan structures and 298 O-glycan structures in GlycoCT format were used to perform the HGI analysis with GlycanFinder. The protein and glycan databases are provided in the Supplementary Data 6. Other search parameters are as follows: EThcD fragmentation for N-glycan analysis and HCD fragmentation for O-glycan analysis; trypsin cleavage, semi-specific digestion with at most one missed cleavage; Cys (Carbamidomethylation, +57.02 Da) as fixed modification, Met (Oxidation, +15.99 Da) and Asn/Gln (Deamidation, +0.98 Da) as variable modifications; precursor error tolerance of 10 ppm, fragment error tolerance of 0.02 Da, glycan fragment error tolerance of 20 ppm.

The search results were further used to calculate eleven different criteria proposed in Kawahara et al.[3] to evaluate the tools' performance. Six criteria N1–N6 were used to evaluate the N-linked glycopeptide performance. The N1 test measures the tools' accuracy to identify glycoPSMs of a synthetic N-glycopeptide that had been included in the dataset as a positive control. The N2 test calculates the Pearson correlation between the expected and the observed N-glycan distribution in human serum. The N3 test measures the accuracy of identifying N-linked glycoproteins expected in human serum. The N4–N6 tests respectively calculate the number of unique N-linked glycopeptides identified, the commonly reported N-linked glycopeptides, and the possible false discoveries containing NeuGc and multi-Fuc that are not expected in human serum. The six criteria were combined and normalized to obtain an overall score from 0 to 1. Similarly, five criteria O1–O5 were used to evaluate the O-linked glycopeptide performance: O-glycan composition (O1), source O-glycoprotein (O2), O-glycoproteome coverage (O3), commonly reported 'consensus' O-glycopeptides (O4), and absence of NeuGc and multi-Fuc O-glycopeptides (O5). A Python script of how to calculate them is provided in the Supplementary Data 6.

**Reporting summary**
Further information on research design is available in the Nature Portfolio Reporting Summary linked to this article.

## Data availability
The fission yeast and mouse datasets from Liu et al.[27] were downloaded from the PRIDE[48] repository with accession numbers: PXD005565 (yeast), PXD005411 (mouse brain), PXD005412 (mouse kidney), PXD005413 (mouse heart), PXD005553 (mouse liver), PXD005555 (mouse lung). The HGI dataset HCD-EThCD-CID-MS/MS (file B) from Kawahara et al.[3] was downloaded from the PRIDE repository with accession number: PXD024101. The IgG datasets generated in our study have been deposited to the PRIDE repository with accession number: PXD039787. Source data are provided with this paper.

## Code availability
The Python implementation, training and testing datasets of the deep learning model for glycan de novo sequencing is available on our GitHub repository GlycoNovo[49] and can be accessed via the following link: https://github.com/zqq66/GlycoNovo. GlycanFinder is available at https://www.bioinfor.com/peaks-studio/, its documentation is provided in the Supplementary Data 7. Our glycan de novo sequencing tool GlycoNovo uses the Python glypy library[50].

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

## Acknowledgements

This work was partially supported by the following grants: National Key R&D Program of China, grant No. 2022YFA1304603 (C.P.), Canada NSERC OGP0046506 (M.L.), the Canada Research Chair program (M.L.), the NSFC No. 61832019 (M.L.), and the Leading Innovative and Entrepreneur teams program of Zhejiang No. 2019R02002 (M.L.).

## Author contributions

W.S., X.Z., L.X., and B.S designed and developed GlycanFinder. Q.Z. and N.H.T. developed the glycan de novo sequencing model. M.Z.R and Z.C. contributed to the software implementation and testing. C.P. and J.M. contributed to the LC-MS/MS experiment. W.S., X.Z., Q.Z., and N.H.T. contributed to the data analysis, result validation, manuscript preparation and revision. M.L., L.X., and B.S supervised the study.

## Competing interests

W.S., X.Z., N.H.T., M.Z.R., Z.C., J.M., L.X., and B.S. are employees of Bioinformatics Solutions Inc. The other authors declare no competing interests.
