## [Peer Review File · Nature Communications]

Glycopeptide database search and de novo sequencing with PEAKS GlycanFinder enable highly sensitive glycoproteomicsREVIEWER COMMENTS

Reviewer #1 (Remarks to the Author):

The manuscript by Sun et al. presented a glycoproteomics software, PEAKS GlycanFinder, which is claimed to provide improved sensitivity and accuracy for N-glycopeptide identification. This is another glycoproteomics tool amongst a rapidly expanding list of other tools recently developed in the field. While GlycanFinder is claimed to be novel by combining database searches and de novo sequencing to increase the glycoproteome coverage, including the identification of novel N-glycans on glycopeptides and correcting for FDRs at glycan and peptide levels, the reviewer was not convinced that this software added significant advancement in the field to warrant publication in Nat Commun. Other software tools have recently demonstrated similar coverage and low FDRs at the glycan and peptide levels and are also able to identify new glycan structures (eg GlycoDecipher, StrucGP, pGlyco 3.0 etc). Disappointingly, the authors also failed to acknowledge these recent developments and appropriately discuss the gaps in the field and how GlycanFinder competes in this space. The manuscript showed many unsupported claims and lacked detailed description of figure legends and methods. For example, the authors claim that this software can identify new glycans, but this was not actually shown by any data. The IgG glycoprofile analysis only shows the performance of the de novo sequence to identify expected glycans. No data were presented regarding the performance of site localization, although described as a feature of this software. Unclear how quantitation of glycopeptides was performed in this software. The results of the "deep profiling of antibody glycosylation" were not compared to any other software. Analysis of IgG glycopeptides on timsTOF (Bruker Daltonics) raised questions whether the improved identification of isomeric peptides was because of the mass spectrometry platform, not the software. The performance comparison with the HGI study was not appropriately conducted and described considering the potential advantages that GlycanFinder had for benchmarking their software after the publication of the 1st HGI study outcome. No method description how the HGI data was searched and analyzed. The authors did not provide the search output of GlycanFinder of all the data used in this study, including the results from IgG and HGI searches. Data presentation was poor and did not meet the high-quality standard of figures for publication in the Nature journals. Finally, the authors neither discussed the shortcoming of their tool nor the limitations of the models used for the performance analysis (eg yeast glycoproteome only contains simple high mannosidic structures which introduced a significant bias in the evaluation and a limitation for the FDR calculation).

Reviewer #2 (Remarks to the Author):

The manuscript describes a new glycoproteomics search engine, GlycanFinder, incorporated into the commercial PEAKS proteomics analysis suite. Methods for analysis of glycoproteomics data are currently progressing rapidly to meet the many challenges of characterizing glycopeptides from mass spectrometry data. The development of new software tools like GlycanFinder is thus very welcome and the manuscript describes several innovations compared to existing methods, notably the de-novo glycan structure search and combined use of peptide- and glycan-based searches. The manuscript is well written and largely contains the information needed to evaluate the performance of GlycanFinder, though some areas require additional details (see below). The inclusion of a comparison to the HGI benchmark study is excellent as well. Major revisions are needed to Figure 2 prior to publication, however, as well as some other smaller revisions throughout.

Major revisions:

- There are several critical problems with Figure 2, stemming from the authors' use of an existing comparison from Zeng et al. (2021) to compare the performance of GlycanFinder to other methods. The desire to use an existing comparison is understandable, however, in this case the comparison of choice was flawed to begin with and is now out of date. There are 3 main issues that need to be addressed in a revised comparison:
 - 1) The analysis by Zeng et al. used different glycan databases for each search engine, meaning the

performance of each tool is not being compared fairly to the others. This would be like comparing proteomics search engines by searching against a full proteome for one engine and an incomplete proteome for another – the results do not provide a valid comparison of the search engines themselves. As an example, Figure 2C highlights a glycosite that was only identified by pGlyco3 and GlycanFinder. This site was only identified with glycans containing 13 or more Hexoses, which were not present in the glycan databases used for the other search engines in Zeng et al.'s analysis (hence why none of the other engines found it). The revised comparison must use the same glycan (and protein) databases for all tools to provide a valid comparison.

2) In addition to lacking NeuAc-containing glycans, the yeast considered here also lacks Fuc-containing glycans. The estimated glycan FDR should be revised to consider both NeuAc- and Fuc-containing glycans as false positives, increasing GlycanFinder's estimated glycan FDR from 0.1% to ~1% (based on a quick calculation using the supplementary tables).

3) Major updates to several of the search tools have been published since Zeng et al.'s analysis was first posted in early 2021, which very much alter the conclusions of the comparison. For example, MSFragger now includes glycan FDR assessment according to a recent publication [Polasky et al. Mol. Cell. Proteomics 2022. <https://doi.org/10.1016/j.mcpro.2022.100205>]. The revised comparison needs to be up to date, either from running up-to-date versions of each tool or taking the updated numbers obtained for the relevant tools in this same dataset from the aforementioned publication.

- A recent publication [Shen, J. et al. Nat. Methods 2021. <https://doi.org/10.1038/s41592-021-01209-0>] described a very similar method, termed StrucGP, which also employs a de-novo glycan structure identification algorithm when a glycan match is not found in the database. This existing method must, at minimum, be discussed and referenced in the text. Ideally, a comparison of the de-novo glycan structure results from GlycanFinder and StrucGP would be provided as well (perhaps added to a revised Figure 2).

- The authors describe GlycanFinder as being able to identify O-linked glycopeptides as well as N-linked, but do not provide any details of how this is done or results showing analysis of O-linked glycosylation in the main text (the HGI benchmark of O-glycopeptide results is only in the supplement). Analysis of O-glycopeptides differs considerably from that of N-linked, making both glycan-based and de-novo methods extremely challenging. Glycan localization is also much more challenging for O-glycopeptides. How GlycanFinder implements these methods for O-glycopeptides and how well they work needs to be addressed in the main text.

- In the analysis of IgG subclasses, FDR from contribution of peptide variant vs glycan composition needs to be addressed, as the difference between IgG1-4 peptides (swapping Phe and Tyr residues) has exactly the same mass difference as swapping Hex and Fuc monosaccharides. The observation of an IgG3 peptide with HexNAc(5)Hex(3)Fuc(1), that was not found in IgG1, 2, or 4, could instead be explained by an IgG2 peptide with HexNAc(5)Hex(4), swapping the position of the +16 mass from Fuc to Phe (in place of Tyr), for example. In spectra without a complete ion series (or with ambiguous fragmentation), how were these possibilities differentiated? An example spectrum with the key distinguishing ion(s) annotated for this case in the supplementary information would be very helpful, as would annotating/highlighting the key distinguishing ions in the spectra shown in Figure 4.

Minor revisions:

- Addition detail is needed in describing the implementation of the peptide-based search: what is the cutoff for searching only the "lower end" of the spectrum for peptide ions? Does that affect sensitivity and FDR if peptides are only scored using a portion of the ions in the spectrum, especially for larger peptides that would tend to have larger fragments?

- It would be very helpful to add a count of how many glycoPSMs were obtained from each of the search strategies (peptide-based, glycan-based, and de-novo) to Figure 2. Additional details regarding how the FDR calculation was performed for glycoPSMs obtained from the de-novo search are also needed (is it different from the database searches? If not, how is an accumulation of false matches in

the de-novo fraction prevented given the larger search space?)

- Figure 1C: typo in "neural network: glycol-PSM" instead of "glyco-PSM"

Reviewer #3 (Remarks to the Author):

The manuscript describes a software program named GlycanFinder that, by combining database search and de novo sequencing techniques, can accurately identify and validate glycopeptides. A deep learning approach is used in particular to construct glycan structures for de novo sequencing. By evaluating how the proposed method performs in comparison to other glycosylation analysis methods, it is shown to be competent.

It is impossible to reproduce the GlycanFinder results presented in the manuscript since the GlycanFinder algorithms, PSM(peptide spectrum match) scoring, and validation procedures are not precisely and thoroughly described. GlycanFinder is an add-on component of the commercial software program PEAKS Studio. I believe the authors would be better off not publishing this manuscript if they did not wish to disclose the specifics of the software. The authors should at the very least make it possible for an anonymous reviewer to use GlycanFinder and PEAKS Studio in order to replicate the results if they don't want to disclose the algorithm's specifics in their manuscript. Even if someone has already purchased PEAKS Studio, they still have to provide their name in order to access GlycanFinder. The training model (together with its hyperparameters) and its learning performance are not at all described in the GitHub repository, despite the fact that the code and datasets for the deep learning model are freely available there.

Reviewer #4 (Remarks to the Author):

In this manuscript, the authors aim to develop a new software tool for extensive glycopeptide identification and quantification. The new approach presented makes use of both glycan as well as peptide fragment assignment and scoring to annotate MS2 data. While such tools are highly required in the field and the weak points of existing packages are well recognized, I don't feel that the current developments make a significant change to existing methods in terms of user friendliness, adaptability and performance. Additionally, not enough detail is described to understand the methods, while fancy (but slightly meaningless) terms, such as "a fine-grained scoring" and "carefully following the target-decoy approach" are used throughout the text to describe performance. Finally, the figure quality is low: There is a lot of unused white spacing, alterations in font size and color, the xy dimensions of the glycan depictions seem skewed and while some panels (Figure 1 a) are hardly readable, others (Figure 1b) take more space than required.

Specific major concerns:

- 1) Throughout the introduction and results, the description of what is actually done remains very vague, exemplified by the following sentence at the end of the introduction: "We performed extensive experiments to evaluate the performance of GlycanFinder, with a special focus on FDR validation and comprehensive benchmarks proposed in a recent community-based evaluation study of the HUPO Human Glycoproteomics Initiative."
- 2) The last sentence of the introduction: "We also obtained the first complete site specific glycosylation profiles of four immunoglobulin G subclasses, overcoming the problem of isomeric peptides and glycans that were challenging to previous studies." is rather bold, how do you define you are the "first complete", when is a study complete? There are hundreds very in-depth IgG glycopeptide studies published, also describing isomers, is this really the most extensive one? To proof that, a proper comparison should be made with existing literature.
- 3) "A glycoPSM passes 1% FDR filter only if both of its peptide and glycan FDRs are less than or equal

1%." This is rather strict, based on RT alignment and accurate mass, glycopeptides can often be assigned based on only glycan fragments.

4) Line 121-130: This description does not contain enough detail to understand. How is "deep learning" defined.

5) Line 169: "Its high sensitivity led to the discovery of new glycopeptides that was previously not possible due to existing limitations in glycoproteomics." This is not specific enough, please discuss what are the existing limitations in glycoproteomics that are solved now.

6) Line 213: "It will be exciting to see future works that can apply deep learning to predict physicochemical properties of glycopeptides, such as fragment ions, retention times, collisional cross sections, etc." This will indeed be the true exciting next step in glycoproteomic data analysis tools, but these steps are not made in the current method, making it unclear what is the added value of the current method.

7) Line 223: "Here we report, for the first time, a deep-coverage and complete profiling of site-specific IgG N linked glycosylation that clearly distinguished the four subclasses IgG1-4, without using any special strategies for sample preparation, labeling, or fragmentation." Please revise these statements about IgG carefully as there are multiple in-depth IgG glycopeptide studies published and no direct comparison was made in this study. Also, not many samples were analyzed here (unclear from how many donors?), making it hard to consider possible biological variation. Additionally, for the Caucasian population it is known that IgG2 and 3 share the same tryptic glycopeptide, making it impossible to distinguish between them. This is not mentioned in the text.

8) The way of quantification is not described. How are multiple charge states and different isotope distributions handled in the quantification?

9) Line 238: It is unexpected that (HexNAc)₄(Hex)₃(Fuc)₁ is the highest abundant glycoform on IgG1 as this goes against all literature. Please validate and discuss this discrepancy.

10) The methods section does not contain enough detail to understand how the software is working. Does it do fragment matching without precursor matching to start with? How much calculation time does this take, and is it feasible to perform these searches on multiple complex samples. Please provide figures on the performance of the method in this regard.

11) Line 321: It is unclear how the complete glycan structure can be defined based on the Y-ions alone.

12) The methods section about "LC-MS/MS analysis of N-linked glycosylation in the four IgG1-4 subclasses" needs revision, both textual as well as content wise. Important details are missing on for example the origin of the used sample and the informed consent provided.

13) Line 440: "IgG glycopeptides were effectively collected." How was it determined that the glycopeptides were effectively collected?

14) Line 445: what is meant by randomly selected in the sentence: "Tryptic IgG glycopeptides was dissolved in solvent A and randomly selected for MS analysis."

15) Line 465: How was the list of 2537 N-linked glycans determined?

Minor concerns:

1) The last sentence of the first paragraph of the introduction is unclear.

2) The term "The tree structures of glycans" is not explained and not often used in the field.

3) Methods: the use of m/z as a unit is not correct. m/z is a quantity, not a unit. The unit is Thomson (Th).

4) Methods: What is the mass accuracy of the oxonium ions considered?

Point-by-point Response Letter

We thank the Reviewers for your constructive comments that greatly helped us to improve our manuscript. We have revised our manuscript and fully addressed all of your concerns. For more details, please see our point-by-point responses to all of your questions on the next pages. Our responses and all revision changes in the manuscript are marked in blue color for your convenience.

Reviewer #1 (Remarks to the Author):

Reviewer #1 - comment 1

Other software tools have recently demonstrated similar coverage and low FDRs at the glycan and peptide levels and are also able to identify new glycan structures (eg GlycoDecipher, StrucGP, pGlyco 3.0 etc). Disappointingly, the authors also failed to acknowledge these recent developments and appropriately discuss the gaps in the field and how GlycanFinder competes in this space.

Authors' response:

We thank Reviewer 1 for pointing this out, it was our mistake not carefully reviewing previous works. In this revised manuscript, we have provided a proper introduction to previous glycan de novo sequencing tools, including StrucGP, Glyco-Decipher, and pGlyco3 (line 114). We discuss the strengths and limitations of those methods, and how our new approach is different from previous works. We have also evaluated our glycan de novo sequencing results against StrucGP on a dataset of five mouse tissues and showed that our deep learning-based approach could learn and predict the tree structures of de novo glycans more accurately (line 186, Figure 3). We also demonstrated that GlycanFinder could identify new glycans that were not found in the database. We hope the new results in this revised manuscript could convince Reviewer 1 of our new contributions to the field.

Reviewer #1 - comment 2

The manuscript showed many unsupported claims and lacked detailed description of figure legends and methods. For example, the authors claim that this software can identify new glycans, but this was not actually shown by any data. The IgG glycoprofile analysis only shows the performance of the de novo sequence to identify expected glycans.

Authors' response:

In this revised manuscript, we have provided the figure legends both at the end of the manuscript and below each figure for your convenience. Our database search and de novo sequencing methods are summarized in the first two sections of the Results section. They are further elaborated in more details in the Methods section. In addition, the details of how we evaluated GlycanFinder and other tools on FDR estimation, HGI benchmarks, and IgG data are also provided in the Methods section.

As mentioned in our response to your previous comment, in this revised manuscript, we have evaluated our glycan de novo sequencing results against StrucGP on a dataset of five mouse tissues and showed that our deep learning-based approach could learn and predict the tree structures of de novo glycans more accurately (line 186, Figure 3). We also demonstrated that GlycanFinder could identify new glycans that were not found in the database.

Reviewer #1 - comment 3

No data were presented regarding the performance of site localization, although described as a feature of this software.

Authors' response:

In this revised manuscript, line 99, we have provided more detailed explanations of the site-specific localization score (A-score) and the structure score (S-score) in GlycanFinder as follows:

“A common problem in glycopeptide analysis is that there may be multiple glycosylation sites in a peptide sequence or multiple glycans with the same composition, which increase the ambiguity of glycoPSM assignment. If the peptide of a glycoPSM has multiple possible glycosylation sites, their localization scores are calculated in GlycanFinder using their respective internal fragment ions. The top-scoring site is then selected and a site-specific localization score, named A-score, is calculated as the score difference between the best and second-best sites. Similarly, when multiple glycans with the same composition match a spectrum, their structure scores are calculated using their respective glycopeptide Y ions. The top-scoring glycan is then selected and an S-score is calculated as the score difference between the best and second-best glycans. The A-score and S-score of a glycoPSM reflect the confidence of its glycosylation site and glycan structure assignments, as larger differences between the best and second-best scores imply higher supporting evidence for the assignments with the best scores.”

The A-score and S-score were calculated for the glycoPSMs of all three datasets, including fission yeast, IgG, and human serum that were used for the performance evaluation of GlycanFinder in this study. They can be found in the Supplementary Tables S1, S3-S6. When comparing our results to other tools such as pGlyco3 or MSFragger, we considered the FDRs, the number of identifications, and the quantifications. Not all tools provide localization scores in their search outputs, so they are not comparable. We also looked at the benchmark criteria in the HGI study by Kawahara et al. and there was no test designed to evaluate the localization.

Reviewer #1 - comment 4

Unclear how quantitation of glycopeptides was performed in this software.

Authors' response:

For each identified glycopeptide-spectrum match (glycoPSM), the area of the isotope feature detected in the LC-MS scan and associated with that MS/MS spectrum is used for quantification of that glycoPSM. The quantification of each glycopeptide is calculated from all of its glycoPSMs. The quantification of each glycan at each glycosylation site on a protein is calculated from all glycopeptides containing that glycan at that site. The section “Antibody N-linked glycosylation profiling” demonstrates how GlycanFinder performed glycopeptide

quantification across four IgG1-4 subclasses and two biological replicates (Figure 4). The quantification results are reported in the column “Area” of the search outputs of GlycanFinder (Supplementary Tables S3-S4).

Reviewer #1 - comment 5, 6

The results of the “deep profiling of antibody glycosylation” were not compared to any other software.

Analysis of IgG glycopeptides on timsTOF (Bruker Daltonics) raised questions whether the improved identification of isomeric peptides was because of the mass spectrometry platform, not the software.

Authors’ response:

We have revised the section “Antibody N-linked glycosylation profiling” to include the analysis results on both Orbitrap (Thermo Fisher Scientific) data and timsTOF (Bruker Daltonics) data obtained from the same IgG sample. We have also compared our results to those of pGlyco3 and MSFragger (Supplementary Figure S5). The search outputs are provided in the Supplementary Tables S3-S4 and the search parameters are described in the Methods section. The results are discussed in the main text, line 226. Below is a summary for your convenience.

“We found that GlycanFinder was able to identify isomeric peptides and glycans on both Orbitrap and timsTOF datasets (Figures 4a, 4b, and Supplementary Figure S4). While timsTOF results contained less number of unique glycopeptides than Orbitrap results (121 and 178, respectively), the top five most abundant glycans and their relative quantifications were consistent on both datasets (Supplementary Figure S4). GlycanFinder, pGlyco3, and MSFragger identified 178, 254, and 88 unique glycopeptides, respectively, from the IgG Orbitrap dataset (Supplementary Figure S5). All glycopeptides of MSFragger were part of GlycanFinder or pGlyco3 identifications. In terms of quantification, pGlyco3 reported the same top five most abundant N-linked glycans and similar quantification profile to that of GlycanFinder (Supplementary Figure S5b). MSFragger results were slightly different, with less amount of (HexNAc)4(Hex)3(Fuc)1 and more of (HexNAc)4(Hex)5(Fuc)1(NeuAc)1. Overall, we observed consistent quantification results from the two different instruments and the three search engines.”

“The LC-MS/MS data was imported into GlycanFinder, pGlyco3, and MSFragger for N-linked glycopeptide analysis. The data was searched against an IgG protein database of 9 entries and an IgG N-linked glycan database of 247 entries. The protein and glycan databases and the raw data have been deposited to the PRIDE repository. The following search parameters were used for all three search engines: HCD fragmentation, trypsin digestion, C(Carbamidomethylation) as fixed modification, M(Oxidation) and NQ(Deamidation) as variable modifications, precursor error tolerance of 10 ppm, fragment error tolerance of 0.02 Da, glycan fragment error tolerance of 20 ppm. The peptide and glycan FDR filters were set at 1%.”

Reviewer #1 - comments 7, 8

The performance comparison with the HGI study was not appropriately conducted and described considering the potential advantages that GlycanFinder had for benchmarking their software after the publication of the 1st HGI study outcome. No method description how the HGI data was searched and analyzed.

The authors did not provide the search output of GlycanFinder of all the data used in this study, including the results from IgG and HGI searches.

Authors' response:

In this revision, we have provided a detailed description of the performance evaluation based on the HGI study (manuscript line 293). In particular, the evaluation was performed on the dataset HCD-EThCD-CID-MS/MS (file B from Kawahara et al., accession number PXD024101). We run both N-linked and O-linked analyses of GlycanFinder, pGlyco3 (build 20210615), and MSFragger (version 19.0) using the same human proteome and glycan databases and the same search parameters, including trypsin cleavage, Cys(Carbamidomethylation) as fixed modification, Met(Oxidation) as variable modification, precursor error tolerance of 10 ppm, fragment error tolerance of 0.05 Da, glycan fragment error tolerance of 20 ppm, peptide and glycan FDRs of 1%.

The search results of GlycanFinder and other search engines are provided in the Supplementary Table S5. They were further used to calculate eleven different criteria proposed in Kawahara et al. to evaluate the tools' performance. The criteria are described in the Supplementary Table S6 and a Python script of how to calculate them is provided in the Supplementary Data 1. For instance, six criteria N1-N6 were used to evaluate N-linked glycopeptide performance. The N1 test measures the tools' accuracy to identify glycoPSMs of a synthetic N-glycopeptide that had been included in the dataset as a positive control. The N2 test calculates the Pearson correlation between the expected and observed N-glycan distribution in human serum. The N3 test measures the accuracy of identifying N-linked glycoproteins expected in human serum. The N4-N6 tests respectively calculate the number of unique N-linked glycopeptides identified, the commonly reported N-linked glycopeptides, and the possible false discoveries containing NeuGc and multi-Fuc that are not expected in human serum. The six criteria were combined and normalized to obtain an overall score from 0 to 1.

We agree with Reviewer #1 that the search results obtained from Kawahara et al. were published about a year before our study and that could be a disadvantage to those tools. We have acknowledged this limitation in the revised manuscript, line 301. We have also tested two other recently published tools, pGlyco3 and MSFragger, using their latest versions and included their results in the performance evaluation above. Overall, the evaluation results show that GlycanFinder performed better than pGlyco3, MSFragger and the other nine softwares from Kawahara et al. on this benchmark dataset (manuscript line 315). We believe the revised analysis here has made the performance evaluation more comprehensive.

The search results of GlycanFinder on the IgG datasets and the comparison to other softwares have also been added in this revision, section “Antibody N-linked glycosylation profiling”. Please see our response to Reviewer #1 - comments 5, 6 above for more details.

Reviewer #1 - comment 9

Data presentation was poor and did not meet the high-quality standard of figures for publication in the Nature journals.

Authors' response:

In this revised manuscript, we have tried to produce the figures with resolution at least 300 dpi and with the sizes required by the journal. We have also provided the figure legends both at the end of the manuscript and below each figure for your convenience. Please let us know which figures that you still find not clear enough and we will try our best to revise them.

Reviewer #1 - comment 10

Finally, the authors neither discussed the shortcoming of their tool nor the limitations of the models used for the performance analysis (eg yeast glycoproteome only contains simple high mannosidic structures which introduced a significant bias in the evaluation and a limitation for the FDR calculation).

Authors' response:

In this revised manuscript, section Discussion, lines 372-390, we have discussed some limitations of our study, including FDR estimation for de novo glycans, the glycoproteome of fission yeast, the limited sample for IgG N-linked glycosylation analysis, and the evaluation benchmarks and results from the HGI study. We hope that these limitations could be addressed in future works.

Reviewer #2 (Remarks to the Author):

Reviewer #2 - comment 1

There are several critical problems with Figure 2, stemming from the authors' use of an existing comparison from Zeng et al. (2021) to compare the performance of GlycanFinder to other methods. The desire to use an existing comparison is understandable, however, in this case the comparison of choice was flawed to begin with and is now out of date. There are 3 main issues that need to be addressed in a revised comparison:

1) The analysis by Zeng et al. used different glycan databases for each search engine, meaning the performance of each tool is not being compared fairly to the others. This would be like comparing proteomics search engines by searching against a full proteome for one engine and an incomplete proteome for another – the results do not provide a valid comparison of the search engines themselves.

As an example, Figure 2C highlights a glycosite that was only identified by pGlyco3 and GlycanFinder. This site was only identified with glycans containing 13 or more Hexoses, which were not present in the glycan databases used for the other search engines in Zeng et al.'s analysis (hence why none of the other engines found it). The revised comparison must use the same glycan (and protein) databases for all tools to provide a valid comparison.

2) In addition to lacking NeuAc-containing glycans, the yeast considered here also lacks Fuc-containing glycans. The estimated glycan FDR should be revised to consider both NeuAc- and Fuc-containing glycans as false positives, increasing GlycanFinder's estimated glycan FDR from 0.1% to ~1% (based on a quick calculation using the supplementary tables).

3) Major updates to several of the search tools have been published since Zeng et al.'s analysis was first posted in early 2021, which very much alter the conclusions of the comparison. For example, MSFragger now includes glycan FDR assessment according to a recent publication [Polasky et al. Mol. Cell. Proteomics 2022. <https://doi.org/10.1016/j.mcpro.2022.100205>].

The revised comparison needs to be up to date, either from running up-to-date versions of each tool or taking the updated numbers obtained for the relevant tools in this same dataset from the aforementioned publication.

Authors' response:

We have followed the reviewer advice and revised this analysis using the latest versions of all tools, searching the same protein and glycan databases for all tools, and considering Fuc-containing glycans as false positives for glycan FDR estimation. The new search results are provided in Supplementary Table S1; the revised analysis is shown in Figure 2 and discussed in the main text, line 158. Below is a summary for your convenience.

“Figure 2a shows that GlycanFinder identified 4,035 glycoPSMs, which was 13.6% more than pGlyco3 (3,553) and 14.5% less than MSFragger (4,720). MetaMorpheus identified the most

number of glycoPSMs (5,232), but it had a highly elevated glycan FDR of 51.6%. The estimated glycan FDRs of GlycanFinder and MSFragger were below 1%, while that of pGlyco3 was 5.0% mainly due to glycans containing Fucose. The peptide FDRs of all tools were well-controlled below 1%. The Venn diagram in Figure 2b shows that GlycanFinder identified 674 glycopeptides, which was 2.7% higher than pGlyco3 (656) and 14.8% less than MSFragger (791).

We also investigated 70 glycopeptides that were identified only by GlycanFinder. Figure 2c shows an example of the glycosylation site N234 on the protein PO13781|YEO3_SCHPO, which had been reported by Zielinska et al.²⁸ using a deglycosylation approach (i.e. only the glycosylation site reported but not the specific glycans). GlycanFinder identified three glycans at this site, including (HexNAc)2(Hex)11 and (HexNAc)2(Hex)13 with high-mannose structures (HexNAc)2(Hex)*n* which are commonly observed in the fission yeast species. Their glycoPSMs with supporting b/y and B/Y ions are shown in Figures 2d and 2e. We also observed that, for the spectrum scan 56805 of sample 2 in this dataset, GlycanFinder identified the high-mannose glycan (HexNAc)2(Hex)11 with better supporting glycopeptide B/Y ions than pGlyco3's identification on this same spectrum (Supplementary Figure S1, Supplementary Table S1). MSFragger did not report any identification at this glycosylation site.”

Overall, the results on this fission yeast dataset shows that GlycanFinder accurately controlled both peptide and glycan FDRs below the expected 1% level. GlycanFinder outperformed pGlyco3 in terms of both FDRs and the number of identifications. GlycanFinder reported 14-15% fewer identifications than MSFragger at similar levels of peptide and glycan FDRs. Each search engine reported extra glycopeptides that were not found by the others.

Reviewer #2 - comment 2

A recent publication [Shen, J. et al. Nat. Methods 2021. <https://doi.org/10.1038/s41592-021-01209-0>] described a very similar method, termed StrucGP, which also employs a de-novo glycan structure identification algorithm when a glycan match is not found in the database. This existing method must, at minimum, be discussed and referenced in the text. Ideally, a comparison of the de-novo glycan structure results from GlycanFinder and StrucGP would be provided as well (perhaps added to a revised Figure 2).

Authors' response:

In this revised manuscript, we have provided a proper introduction to previous glycan de novo sequencing tools, including StrucGP, Glyco-Decipher, and pGlyco3 (line 114). We discuss the strengths and limitations of those methods, and how our new approach is different from previous works. We have also evaluated our glycan de novo sequencing results against StrucGP on a dataset of five mouse tissues and showed that our deep learning-based approach could learn and predict the tree structures of de novo glycans more accurately (line 186, Figure 3). We also demonstrated that GlycanFinder could identify new glycans that were not found in the database.

Reviewer #2 - comment 3

The authors describe GlycanFinder as being able to identify O-linked glycopeptides as well as N-linked, but do not provide any details of how this is done or results showing analysis of O-linked glycosylation in the main text (the HGI benchmark of O-glycopeptide results is only in the supplement). Analysis of O-glycopeptides differs considerably from that of N-linked, making both glycan-based and de-novo methods extremely challenging. Glycan localization is also much more challenging for O-glycopeptides. How GlycanFinder implements these methods for O-glycopeptides and how well they work needs to be addressed in the main text.

Authors' response:

We have added a new section, "Evaluation of O-linked glycopeptide analysis", in the revised manuscript, line 329, and new Figure 6 and Supplementary Figure S6 to provide the key points of O-linked glycopeptide analysis in GlycanFinder. We have also evaluated the performance of O-linked glycopeptide database search engines, including GlycanFinder, pGlyco3, and MSFragger, based on the HGI benchmarks.

Our de novo sequencing is currently only applied to N-linked glycans, as we found it much more challenging to O-linked glycans. The O-linked search space is more complicated due to multiple possible occurrences of O-linked glycans per peptide. Furthermore, O-linked glycans appear to have less predictable structures than N-linked and their existing databases are very limited, making it difficult for machine learning models to learn useful features to assist de novo sequencing. Thus, de novo sequencing of O-linked glycans still remains a challenging problem for future research topics, and currently there has been no solution proposed to address this problem (to the best of our knowledge). We have acknowledged this limitation in the Discussion of the revised manuscript, line 377.

The section "Evaluation of O-linked glycopeptide analysis" is copied below for your convenience:

"GlycanFinder also performs peptide-based and glycan-based searches of O-linked glycopeptides in a similar fashion to that of N-linked glycopeptides. However, unlike N-linked glycopeptides which usually have a N-X-S/T/C (X≠P) motif, O-linked glycans are attached to proteins via the hydroxyl groups of serine (S) or threonine (T) residues. Thus, there are often more than one O-glycosylation sites in a peptide sequence. GlycanFinder allows at most two O-linked glycans per peptide and considers internal fragment ions to determine the best glycosylation sites and to calculate the site-specific localization score (A-score) to reflect the confidence of its assignments (Supplementary Figure S6). The benchmark results for O-linked glycopeptides are shown in Figure 6. GlycanFinder achieved a higher overall score than pGlyco3 and MSFragger (0.730, 0.431, and 0.654, respectively). GlycanFinder also outperformed the other nine softwares as well as the best result (user team 13) reported in Kawahara et al.³. GlycanFinder performed consistently well across four criteria O2-O5 (0.780-1.000) while having a low score for criterion O1 (O-glycan composition). All other tools except IQ-GPA also scored low on this criterion O1, which measures the Pearson correlation between the expected and the observed O-glycan distribution in human serum. The same

results for O1 had also been observed in Kawahara et al.³, where using a narrow search space and permitting only few missed peptide cleavages were suggested to improve O1 score. Overall, the benchmark results based on this community evaluation study demonstrate that GlycanFinder represents a high-performance informatics solution for both N-glycoproteomics and O-glycoproteomics³.”

Reviewer #2 - comment 4

In the analysis of IgG subclasses, FDR from contribution of peptide variant vs glycan composition needs to be addressed, as the difference between IgG1-4 peptides (swapping Phe and Tyr residues) has exactly the same mass difference as swapping Hex and Fuc monosaccharides.

...An example spectrum with the key distinguishing ion(s) annotated for this case in the supplementary information would be very helpful, as would annotating/highlighting the key distinguishing ions in the spectra shown in Figure 4.

Authors' response:

We have followed the reviewer's advice to provide detailed annotations of the key ions that could distinguish peptides and glycans of IgG1-4 in Figure 4. In particular, the glycopeptide B/Y ions with higher masses and intensities allowed to identify the glycan structure (Figure 4a), whereas the peptide backbone b/y ions with lower masses and intensities helped to identify the peptide sequences (Figure 4b). The key ions b3/b4, y5/y6, b7/b8, and y1/y2 corresponding to Tyr (Y) and Phe(F) amino acids at the 4th and 8th positions allowed to distinguish the four glycopeptides of IgG1-4 (Figure 4b and Supplementary Figure S3). In addition to the analysis results of the IgG Orbitrap dataset in Figure 4, we also performed the same analysis on the IgG timsTOF dataset and found that isomeric glycopeptides of IgG1-4 could also be differentiated in the timsTOF results (Supplementary Figure S4).

The peptide and glycan FDRs in GlycanFinder are estimated using a target-decoy approach. For peptide FDR, the decoy protein database is generated by randomly shuffling the target protein database. For glycan FDR, we apply random mass shifts of fragment ions to create decoy spectra, as previously described in Fang et al²⁰. A glycoPSM passes 1% FDR filter only if both of its peptide and glycan FDRs are less than or equal 1%. Thus, the FDRs are controlled globally for any general datasets. Currently GlycanFinder does not apply a specific FDR calculation to handle the swapping of Phe and Tyr residues and the swapping of Hex and Fuc monosaccharides for the IgG analysis. We are also not aware of any other search engines that can handle such cases.

Reviewer #2 - comment 5

It would be very helpful to add a count of how many glycoPSMs were obtained from each of the search strategies (peptide-based, glycan-based, and de-novo) to Figure 2. Additional details regarding how the FDR calculation was performed for glycoPSMs obtained from the de-novo search are also needed (is it different from the database searches? If not, how is an accumulation of false matches in the de-novo fraction prevented given the larger search space?)

Authors' response:

We did not find any glycoPSMs with de novo glycans from the fission yeast dataset. Probably because the fission yeast is a well-studied model organism with a relatively simple glycosylation profile. Thus, it is more suitable for FDR validation. It may be more interesting to look for de novo glycans in more complicated datasets, such as the mouse dataset in the section "Evaluation of glycan de novo sequencing".

Certainly it is very important to estimate the FDRs of glycoPSMs identified from the de novo search. A proper way to do this is to add the de novo glycans to the original glycan database and then repeat the glycopeptide database search with the new glycan database and FDR control. This approach has been demonstrated successfully for HLA peptide de novo sequencing (Tran et al., Nature Machine Intelligence, 2020. <https://doi.org/10.1038/s42256-020-00260-4>). Unfortunately, this function has not been implemented in GlycanFinder yet and we plan to include it in a future release. We have acknowledged this limitation in the Discussion section, line 372.

Most of the 4,035 glycoPSMs of GlycanFinder in Figure 2 were identified during the peptide-based search. 94 (2.3%) glycoPSMs were not identified in the peptide-based search and were subsequently discovered in the glycan-based search.

Reviewer #3 (Remarks to the Author):

Reviewer #3 - comment 1

It is impossible to reproduce the GlycanFinder results presented in the manuscript since the GlycanFinder algorithms, PSM(peptide spectrum match) scoring, and validation procedures are not precisely and thoroughly described.

Authors' response:

In this revised manuscript, we have provided very detailed descriptions of our methods and analysis results. In particular, our database search and de novo sequencing methods are summarized in the first two sections of the Results section. They are further elaborated in more details in the Methods section. A full documentation of the GlycanFinder software is also available in the Supplementary Data 2. The search outputs (glycoPSMs) of GlycanFinder and other tools are provided in the following Supplementary Tables: S1 (fission yeast), S2 (five mouse tissues), S3-S4 (IgG datasets), and S5-S6 (HGI study). In addition, the details of how we evaluated GlycanFinder and other tools on FDR estimation, HGI benchmarks, and IgG data are also provided in the Methods section. Please let us know which section that you still find not clear enough and we shall try to provide more explanations.

Reviewer #3 - comments 2, 3

GlycanFinder is an add-on component of the commercial software program PEAKS Studio. I believe the authors would be better off not publishing this manuscript if they did not wish to disclose the specifics of the software. The authors should at the very least make it possible for an anonymous reviewer to use GlycanFinder and PEAKS Studio in order to replicate the results if they don't want to disclose the algorithm's specifics in their manuscript. Even if someone has already purchased PEAKS Studio, they still have to provide their name in order to access GlycanFinder.

The training model (together with its hyperparameters) and its learning performance are not at all described in the GitHub repository, despite the fact that the code and datasets for the deep learning model are freely available there.

Authors' response:

In this revision, we have provided a download link and accounts in the cover letter to the Editor so that the Editor and the Reviewers can use GlycanFinder and validate the results. We have also provided the new Python implementation, pretrained models, training and testing datasets of the deep learning model for glycan de novo sequencing on GitHub via the following new link: <https://github.com/zqq66/GlycoNovo>.

Reviewer #4 (Remarks to the Author):

Reviewer #4 - comment 1

Additionally, not enough detail is described to understand the methods, while fancy (but slightly meaningless) terms, such as “a fine-grained scoring” and “carefully following the target-decoy approach” are used throughout the text to describe performance.

Authors' response:

In this revised manuscript, we have provided very detailed descriptions of our methods and analysis results. In particular, our database search and de novo sequencing methods are summarized in the first two sections of the Results section. They are further elaborated in more details in the Methods section. The search outputs (glycoPSMs) of GlycanFinder and other tools are provided in the following Supplementary Tables: S1 (fission yeast), S2 (five mouse tissues), S3-S4 (IgG datasets), and S5-S6 (HGI study). In addition, the details of how we evaluated GlycanFinder and other tools on FDR estimation, HGI benchmarks, and IgG data are also provided in the Methods section. Please let us know which section that you still find not clear enough and we shall try to provide more explanations.

We have also removed the terms that sound fancy or meaningless as the reviewer suggested.

Reviewer #4 - comment 2

Finally, the figure quality is low: There is a lot of unused white spacing, alterations in font size and color, the xy dimensions of the glycan depictions seem skewed and while some panels (Figure 1 a) are hardly readable, others (Figure 1b) take more space than required.

Authors' response:

We have revised Figure 1 to reduce unused white spacing as the reviewer suggested. In this revised manuscript, we have tried to produce the figures with resolution at least 300 dpi and with the sizes required by the journal. We have also provided the figure legends both at the end of the manuscript and below each figure for your convenience. Please let us know which figures that you still find not clear enough and we will try our best to revise them.

Reviewer #4 - comment 3

Throughout the introduction and results, the description of what is actually done remains very vague, exemplified by the following sentence at the end of the introduction: “We performed extensive experiments to evaluate the performance of GlycanFinder, with a special focus on FDR validation and comprehensive benchmarks proposed in a recent community-based evaluation study of the HUPO Human Glycoproteomics Initiative.”

Authors' response:

We have removed unnecessary terms as mentioned earlier and revised the sentence as follows:

“We performed experiments to evaluate the performance of GlycanFinder, including FDR validation and benchmarks proposed in a recent community-based evaluation study of the HUPO Human Glycoproteomics Initiative.”

The sentence basically summarizes the evaluations that we did (FDR, HGI benchmarks) and the details are then described in the section Results.

Reviewer #4 - comment 4

The last sentence of the introduction: “We also obtained the first complete site specific glycosylation profiles of four immunoglobulin G subclasses, overcoming the problem of isomeric peptides and glycans that were challenging to previous studies.” is rather bold, how do you define you are the “first complete”, when is a study complete? There are hundreds very in-depth IgG glycopeptide studies published, also describing isomers, is this really the most extensive one? To proof that, a proper comparison should be made with existing literature.

Authors' response:

We have removed the words “first complete” and revised the sentence as follows (line 75):

“We also demonstrated an N-linked glycosylation profiling of four immunoglobulin G subclasses that could distinguish isomeric peptides and glycans, which had been a challenging problem to previous studies^{18–20}.”

We have also removed other words and statements such as “deep”, “complete”, “first”, etc. throughout the manuscript that do not have proper supporting or comparison evidence.

Reviewer #4 - comment 5

“A glycoPSM passes 1% FDR filter only if both of its peptide and glycan FDRs are less than or equal 1%.” This is rather strict, based on RT alignment and accurate mass, glycopeptides can often be assigned based on only glycan fragments.

Authors' response:

We found that some other recent studies, such as pGlyco3 and MSFragger, also applied both peptide and glycan FDRs. Below are the references for your convenience. We believe that controlling both peptide and glycan FDRs is essential as glycopeptide spectra are complex and contain multiple types of fragment ions. There are several amino acid residues and monosaccharides or their combinations that have similar or identical masses and thus may increase the chance of random matches and the number of false identifications.

pGlyco3: <https://doi.org/10.1038/s41592-021-01306-0>

MSFragger: <https://doi.org/10.1016/j.mcpro.2022.100205>

Reviewer #4 - comments 6, 7

Line 169: "Its high sensitivity led to the discovery of new glycopeptides that was previously not possible due to existing limitations in glycoproteomics." This is not specific enough, please discuss what are the existing limitations in glycoproteomics that are solved now.

Line 213: "It will be exciting to see future works that can apply deep learning to predict physicochemical properties of glycopeptides, such as fragment ions, retention times, collisional cross sections, etc." This will indeed be the true exiting next step in glycoproteomic data analysis tools, but these steps are not made in the current method, making it unclear what is the added value of the current method.

Authors' response:

As stated in the abstract, the new contributions presented in our study include a new glycopeptide database search that integrates both peptide-based and glycan-based strategies, a new glycan de novo sequencing model, a new antibody N-linked glycosylation profiling that could distinguish isomeric peptides and glycans in four immunoglobulin G subclasses. We have also evaluated our proposed methods against other existing tools on multiple datasets, including the fission yeast, five mouse tissues, human IgG, and human serum. We hope the results presented in this revised manuscript could convince the reviewer of our new contributions to the field.

Reviewer #4 - comment 8

Line 223: "Here we report, for the first time, a deep-coverage and complete profiling of site-specific IgG N linked glycosylation that clearly distinguished the four subclasses IgG1-4, without using any special strategies for sample preparation, labeling, or fragmentation."

Please revise these statements about IgG carefully as there are multiple in-depth IgG glycopeptide studies published and no direct comparison was made in this study.

Also, not many samples were analyzed here (unclear from how many donors?), making it hard to consider possible biological variation. Additionally, for the Caucasian population it is known that IgG2 and 3 share the same tryptic glycopeptide, making it impossible to distinguish between them. This is not mentioned in the text.

Authors' response:

We have revised the sentence as follows:

“Here we reported an N-linked glycosylation profiling that could distinguish the four subclasses IgG1-4, without using any special strategies for sample preparation, labeling, or fragmentation¹⁸.”

We have also removed other words and statements such as “deep”, “complete”, “first”, etc. throughout the manuscript that do not have proper supporting or comparison evidence.

We thank the reviewer for the information about the Caucasian population. We have added that information in the revised manuscript, line 248 as follows:

“It should be noted that, for the Caucasian population, IgG2 and IgG3 share the same tryptic N-linked glycopeptides, making it impossible to distinguish them^{32,33}.”

In this revised study, we used a commercial IgG sample which was purchased from Solarbio Life Sciences (catalog SP001). We agree with the reviewer that the limited sample was not enough to account for biological variation. We have acknowledged this limitation in the Discussion, line 386.

Reviewer #4 - comment 9

The way of quantification is not described. How are multiple charge states and different isotope distributions handled in the quantification?

Authors' response:

For each identified glycopeptide-spectrum match (glycoPSM), the area of the isotope feature detected in the LC-MS scan and associated with that MS/MS spectrum is used for quantification of that glycoPSM. The quantification of each glycopeptide is calculated from all of its glycoPSMs. The quantification of each glycan at each glycosylation site on a protein is calculated from all glycopeptides containing that glycan at that site. The section “Antibody N-linked glycosylation profiling” demonstrates how GlycanFinder performed glycopeptide quantification across four IgG1-4 subclasses and two biological replicates (Figure 4). The quantification results are reported in the column “Area” of the search outputs of GlycanFinder (Supplementary Tables S3-S4).

Reviewer #4 - comment 10

Line 238: It is unexpected that (HexNAc)₄(Hex)₃(Fuc)₁ is the highest abundant glycoform on IgG1 as this goes against all literature. Please validate and discuss this discrepancy.

Authors' response:

In this revision, we have repeated the quantification of IgG N-linked glycans on both Orbitrap and timsTOF datasets and we have also compared our results to two other softwares, pGlyco3 and MSFragger (revised manuscript, line 230). We observed consistent quantification results

from the two different instruments and the three search engines (Figure 4, Supplementary Figures S4-S5). In particular, the top five most abundant glycans on IgG1 and IgG2 are: (HexNAc)4(Hex)4(Fuc)1, (HexNAc)4(Hex)5(Fuc)1, (HexNAc)4(Hex)3(Fuc)1, (HexNAc)4(Hex)5(Fuc)1(NeuAc)1, and (HexNAc)5(Hex)4(Fuc)1. The glycan intensities on IgG1 and IgG2 were found to be 2-3 orders of magnitudes higher than on IgG3 and IgG4. Thus, it is probable that the glycosylation profiles observed here were related to the source of the sample under investigation. We have added this discussion in the revised manuscript, line 262.

Reviewer #4 - comment 11

The methods section does not contain enough detail to understand how the software is working. Does it do fragment matching without precursor matching to start with? How much calculation time does this take, and is it feasible to perform these searches on multiple complex samples. Please provide figures on the performance of the method in this regard.

Authors' response:

In this revised manuscript, we have provided very detailed descriptions of our methods and analysis results. In particular, our database search and de novo sequencing methods are summarized in the first two sections of the Results section. They are further elaborated in more details in the Methods section. A full documentation of the GlycanFinder software is also available in the Supplementary Data 2 to provide step-by-step instructions of how the software works, how to interpret the results, and other performance-related issues such as configurations, memory, running time, etc.

Reviewer #4 - comment 12

The methods section about “LC-MS/MS analysis of N-linked glycosylation in the four IgG1-4 subclasses” needs revision, both textual as well as content wise. Important details are missing on for example the origin of the used sample and the informed consent provided.

Line 465: How was the list of 2537 N-linked glycans determined?

Authors' response:

As mentioned earlier, in this revised study, we used a commercial IgG sample which was purchased from Solarbio Life Sciences (catalog SP001). We have added that information in the manuscript, line 526.

We have also revised the IgG LC-MS/MS data analysis (manuscript, line 580) as follows:

LC-MS/MS data analysis.

The LC-MS/MS data was imported into GlycanFinder, pGlyco3, and MSFragger for N-linked glycopeptide analysis. The data was searched against an IgG protein database of 9 entries and an IgG N-linked glycan database of 247 entries. The protein and glycan databases and the raw data have been deposited to the PRIDE repository. The following search parameters were used for all three search engines: HCD fragmentation, trypsin digestion, C(Carbamidomethylation) as fixed modification, M(Oxidation) and NQ(Deamidation) as variable modifications, precursor error tolerance of 10 ppm, fragment error tolerance of 0.02 Da, glycan fragment error tolerance of 20 ppm. The peptide and glycan FDR filters were set at 1%.

The IgG protein and glycan databases were determined based on our domain knowledge of those proteins and glycans that are commonly associated with human IgG.

REVIEWER COMMENTS

Reviewer #1 (Remarks to the Author):

In the revised manuscript, the authors have addressed most of the reviewer's comments and performed additional analysis that improved the quality of the manuscript. The authors have also provided the detailed list of identifications from all the searches performed in their study. PEAKS GlycanFinder shows a good potential to be competitive and within the top performance glycoproteomics software. However, for publication purpose, the quality of the figures remains poor. The legend of the panels (eg, a, b, c) needs to be aligned to the left. There are many graphs lacking legend in the y-axis (eg Fig 3a, Fig 4d etc). The annotated spectra and glycan cartoons of Fig 3 and 4 are too small. If glycan cartoons are displayed, this needs to come with glycan key. The style of the figure panels is not consistent, some are displayed with borders, others without. Figure 2c seems to be a print screen of output report of the software and some of the details do not provide any meaningful information for the reader (eg AA per line, deltaM, H, #). Legend of Figure 3 lack detail description (eg, how accuracy was determined in panel a?). When graphs are showing quantification, this needs to be clearly described in the legend how this was performed (eg figure 4c no info of the quantification; the authors used the term "intensity" in figure 4d but in the rebuttal letter and supplementary tables they used the term "area"). How was normalization of the glycopeptide area performed? The site specific glycoform distribution is better represented by normalized area within each site not absolute "area". The quantification method needs to be clearly described in the method section. Please describe how the HGI dataset was searched in the methods section. O-glycopeptide report Identification is missing in the Supplementary Table S5. Supplementary Table S6 is just a copy of the information already provided in Kawahara et al and should be only referenced but not shown as supplementary. Fig 6 is better placed as supplementary since the other analysis focused on N-glycopeptide only. MSFragger has been benchmarked by the HGI study and demonstrated better performance compared to what the authors showed in Figure 5 (<https://www.hupo.org/page-1757445>). Likewise, the authors should consider benchmarking their software against the HGI study, a service they appear to offer, to avoid any bias or incorrect use of the score system. If the authors want to show a comparison against the HGI study it is recommended that only direct measurements are used (eg the authors could show a comparison of number of glycoPSMs reported by the HGI teams and PEAKS GlycanFinder, glycan composition correlation against a reference serum glycome, % of NeuGc and multi Fuc).

Reviewer #2 (Remarks to the Author):

Sun et al. have revised the manuscript describing their GlycanFinder software package, rerunning the comparison of the yeast FDR experiment and HGI benchmarking analysis and adding many more details about the method and IgG analysis. These revisions have greatly improved the paper and satisfied most of the issues I saw in the original manuscript. However, changes in some areas have raised additional issues that need to be resolved.

Major issues:

- I originally praised the authors for including the HGI benchmark for their software tool, and the addition of more detail on how the benchmark was performed is very welcome. However, the inclusion of the authors' attempts to run several other software tools and score them with the benchmark to show that GlycanFinder performed better is problematic and should be removed. In the HGI study, each software tool was run either by its own developers, or users of the software. In either case, the group submitting results was motivated to provide the best results possible, and a neutral third party assessed the performance. Notably, no developers ran another developer's software tool, because they would have a conflict/bias in wanting their software to perform the best (regardless of whether this was acted on). I do not mean to imply that the authors intended to bias the analysis, however, because this deviates considerably from how the actual benchmark was done, the other software tools

analyzed by the GlycanFinder team should not be included in the benchmark figure.

- Additionally, insufficient detail is still provided to understand the HGI analysis even for GlycanFinder. What glycan database was used? The text states that all tools used the same database but not what that database was. This is crucial as using the final database obtained by the HGI study would be a significant (and potentially unfair) advantage for GlycanFinder over the other tools previously run in the benchmark.

- The authors have provided additional details on how O-glycopeptides are analyzed, but some details remain unclear and need to be provided, and they have neglected to mention several key limitations.
 - o The authors state that GlycanFinder uses "internal" fragment ions to determine the location of O-glycans within the peptide, but it appears from supporting figure 6 that only the standard terminal ions are used (b/y and c/z). Internal ions are multiply fragmented sections of the peptide that contain neither terminus, resulting from a combination of b- and y-type fragmentation events (for example). These ions do exist in peptide fragmentation spectra at low levels, but they are unlikely to be of use in localization in most cases and have a significant potential for false matching given their much larger search space. I believe the authors did not mean to refer to internal ions, so please clarify if this is case. If the authors do mean internal ions, how specifically they are being used (which ions are used? by and/or ay? cy, bz, cz? How often are they actually observed?) and what steps are being taken to account for the large search space and potential for false localization need to be addressed.
 - o No discussion of activation method is provided, despite O-glycans generally requiring ETD (or similar) for localization. How are O-glycans localized in HCD scans by GlycanFinder? Figure S6 only shows EThcD scans. Or if they cannot be localized in HCD scans, what is reported for HCD scans searched for O-glycopeptides? How was this handled for the HGI benchmark?
 - o The ion types used for localization are not listed – are b/y ions used for localization? If so, what part of the glycan is being retained on these ions that can be used to localize (please show an example spectrum)? b/y+HexNAc fragments are quite rare for O-glycans (unlike N-glycans). If only c/z ions are used, please state that.
 - o The limitation of max 2 glycans per peptide is significant, as many O-glycopeptides contain more than 2 glycans. Many of the HGI benchmark study O-glycopeptides contained >2 glycans – how were these reported in the results, and how is GlycanFinder able to obtain such high scores despite not handling these glycopeptides?

Minor Issues:

- The comparison to StrucGP in Figure 3 is a welcome addition to the paper, and appears to show good performance for GlycanFinder as some de novo glycans are found that are missed by StrucGP. However, it is stated in the Discussion that the FDR for de novo glycans is not assessed. How were these discovered de novo glycans compared to StrucGP in a fair way? Either the FDR is not assessed (unlike StrucGP) and it is unclear if the additional glycans are correct or not, or the FDR is assessed in a second follow-up run. If the latter, this needs to be stated in the text so readers are not misled about how the analysis is performed and how it differs from the single run of StrucGP.

- In both the abstract and introduction, the authors state that "Our results show that GlycanFinder not only controlled the FDRs accurately but also identified more glycopeptide-spectrum matches (glycoPSMs) than previously reported." However, Fig 2 clearly shows GlycanFinder identifying fewer glycoPSMs than competing tools. Please adjust these statements to accurately reflect the data shown in the manuscript.

- Page 4, line 70 – the authors state "GlycanFinder accurately estimates the FDRs at both peptide and glycan levels by ... designing proper decoys for those levels." While the mass shifted glycan FDR is the best method developed thus far, it is not a proper decoy as it significantly underestimates the rate of false matches to glycans with shared fragment ions (i.e., most glycans). Please note this general limitation of this glycan FDR method.

- Page 8, line 191 – which of the 5 mouse tissues was the one used for testing (vs the other 4 for training)? Are the results in Fig. 3 shown from the training or testing data (or both)?

Reviewer #3 (Remarks to the Author):

The manuscript describes GlycanFinder, a software program that can accurately identify and validate glycopeptides by combining database search and de novo sequencing techniques. When a glycopeptide cannot be determined from a database search, a deep learning approach is used to reconstruct glycan structures through de novo sequencing.

Major comments:

1. The performance of GlycanFinder is somewhat disappointing. Database search results in Figure 2 showed that MSFragger resulted in 17% more glycoPSMs when FDR was estimated at 1%. For the IgG dataset analysis, pGlyco3 identified 254 unique glycopeptides, while GlycanFinder identified 178. De novo glycan sequencing results shown in Figure 3 gave better performance than StrucGP but its structural accuracy was just above 30%. Given that this performance was obtained through machine learning, it is difficult to believe that the training was successful.

2. In lines 197-203, the accuracy comparison between GlycanFinder and StrucGP was made and Supplementary Table S2 was referred to. However, the table provides identification results that cannot be directly compared.

3. In lines 219-220, it is stated that “the accuracy of glycan de novo sequencing was on par with that of de novo peptide sequencing”. What grounds did you use to make such a claim? Glycan de novo sequencing accuracy was slightly higher than 30%, significantly lower than the usual de novo peptide sequencing accuracy.

4. Given that the performance of GlycanFinder in DB search is not superior to the other existing methods, the main contribution of the manuscript is the de novo glycan sequencing part. However, as I previously stated in my review comment, the description of the training model is only at a very abstract level, with insufficient details. It must include (1) model architecture (how many layers, how many units, any residual blocks, etc.) (2) the input representation and its dimension (3) hyperparameters (4) the final size of learnable parameters(weights) (5) training vs. validation errors during training, at the very least.

5. While the authors emphasized the importance of accurate FDR estimation, they did not describe how it is done, omitting how peptide and glycan scores are calculated. In the first place, it must be precisely defined, and the score distributions between target and decoy PSMs must be presented.

6. In the same vein, it is not described how A-score is computed by GlycanFinder. It is vaguely described as “comparing glycosylation sites and their respective internal fragment ions”. By the way, how do you define “internal fragment ion” in a glycoPSM?

7. Contrary to the original purpose of A-score for PTM site localization, it appears that A-score is simply included as an output of GlycanFinder and is not used to confidently localize glycosylation sites. Why?

8. When estimating an FDR for IgG data, the database consisted of 9 protein sequences and 247 glycans. What statistical significance do you believe a target-decoy validation has when such a tiny database is used?

9. It is not clear how a glycan is sequenced de novo. According to Figure 1c, a monosaccharide appears to be selected and it can only be attached to a leaf node, not an internal node of a tree. If

that's the case, I am not sure if we can call it a "true" de novo glycan sequencing. If not, it must be clearly stated to what extent novel glycan structures can be deduced from tandem mass spectra.

10. In Figure 4, the authors emphasized the challenge of distinguishing the four different isomeric glycopeptides of IgG1-4. It is not clear why this is a challenge. The four peptides from IgG1 vs. IgG2 vs. IgG3/IgG4 have different parent masses. Any algorithm could distinguish between IgG3 and IgG4 if their backbone fragmentations were successful, as shown in Fig. 4b.

Minor comment:

1. Figure 4b: for both mass spectra of peptides "EEQYNSTFR" and "EEQFNSTYR" at the bottom panel, two fragment ions flanking the annotated amino acid in blue letters (F and Y, respectively) are not clearly shown.

Reviewer #4 (Remarks to the Author):

Although some of the comments were addressed, the authors were not convincing in conveying that the current developments make a significant change to existing methods in terms of user friendliness, adaptability and performance, additionally the representation of the work is not sufficient.

For example, what are the improvements when compared to recently published tools in Nature Communications, such as pGlycoQuant (<https://www.nature.com/articles/s41467-022-35172-x>)? This tool offers solutions to most of the described issues, and more, including MS1-based identification and quantification. The comparison between the current tool and pGlyco3 and MSFragger shows similar performance, it is unclear what is the advantage of the current tool.

Still, the figure quality is low: There is a lot of unused white spacing, alterations in font size and color. Most graphs are directly copied from excel, without careful formatting, some mass spectra are so small that they are unreadable. The figures do not meet the high-quality standard of figures for publication in the Nature journals.

It is still unclear how quantification is performed. What do the authors mean by "the area of the isotope feature detected in the LC-MS scan and associated with that MS/MS spectrum is used for quantification of that glycoPSM. The quantification of each glycopeptide is calculated from all of its glycoPSMs". I do not expect glycoPSMs to be quantified individually, rather used for the identification of glycopeptides. The relevant MS1 traces of the glycopeptide (including relevant isotopes and charge states) should then be used for quantification. Please describe how this is approached, or if a completely different approach is taken.

It is important to discuss memory and run time in the main text and compare this to other tools.

It is unclear how "This approach guarantees that we do not miss any candidates, as opposed to using one strategy alone". How can you prove that no candidates are missed?

It is unclear how glycan and peptide score are determined, what calculations are made to come to these scores?

Point-by-point Response Letter

We thank the Reviewers for your constructive comments that greatly helped us to improve our manuscript. We have revised our manuscript and fully addressed all of your concerns. For more details, please see our point-by-point responses to all of your questions on the next pages. Our responses and all revision changes in the manuscript are marked in blue color for your convenience.

Reviewer #1 (Remarks to the Author):

In the revised manuscript, the authors have addressed most of the reviewer's comments and performed additional analysis that improved the quality of the manuscript. The authors have also provided the detailed list of identifications from all the searches performed in their study. PEAKS GlycanFinder shows a good potential to be competitive and within the top performance glycoproteomics software.

Reviewer #1 - comment 1

However, for publication purpose, the quality of the figures remains poor. The legend of the panels (eg, a, b, c) needs to be aligned to the left. There are many graphs lacking legend in the y-axis (eg Fig 3a, Fig 4d etc). The annotated spectra and glycan cartoons of Fig 3 and 4 are too small. If glycan cartoons are displayed, this needs to come with glycan key. The style of the figure panels is not consistent, some are displayed with borders, others without. Figure 2c seems to be a print screen of output report of the software and some of the details do not provide any meaningful information for the reader (eg AA per line, deltaM, H, #). Legend of Figure 3 lack detail description (eg, how accuracy was determined in panel a?).

Authors' response:

In this revised manuscript, we have improved the quality of the figures as Reviewer #1 suggested. In particular, the panel legends (a, b, c, etc) have been aligned to the left. The y-axes of all charts have been fully annotated. We have doubled the sizes of the annotated spectra and glycans in Figures 3 and 4 so that they can be clearly displayed. The spectra are also annotated with their scan numbers, which can be used to look up the accompanied supplementary files for more details of the respective glycoPSMs, including the glycan ID, structure, identification score, etc. The glycan compositions are also displayed next to the glycan cartoons. We have removed the borders of all figure panels for consistency. We have also removed Figure 2c as it does not provide meaningful information. We have added the description of the accuracy calculation in the legend of Figure 3; we have also provided more detailed descriptions in the legends of the other figures.

Reviewer #1 - comment 2

When graphs are showing quantification, this needs to be clearly described in the legend how this was performed (eg figure 4c no info of the quantification; the authors used the term "intensity" in figure 4d but in the rebuttal letter and supplementary tables they used the term "area"). How was normalization of the glycopeptide area performed? The site specific glycoform distribution is better represented by normalized area within each site not absolute "area". The quantification method needs to be clearly described in the method section.

Authors' response:

We have added a section named “GlycanFinder quantification of glycopeptides” in the Methods section, line 588. We copied it below for your convenience.

“GlycanFinder quantification of glycopeptides

For each identified glycopeptide, the sum of the areas of its MS1 precursor features with different charges is used for its quantification. The label-free approach is applied for glycopeptide quantification among different samples. Given a glycosylation site of a glycoprotein, the glycan profile is calculated based on the normalized areas of the glycan forms at that site. First, the area of each glycan form at each glycosylation site is calculated by summing the areas of all glycopeptides containing that glycan at that site. Then, the normalized area of each glycan form is obtained by dividing its area by the sum of the areas of all glycans forms at that site.”

We have also revised Figure 4 to use the normalized areas to represent the site-specific glycoform distribution as Reviewer #1 suggested.

Reviewer #1 - comment 3

Please describe how the HGI dataset was searched in the methods section.

Authors' response:

We have added the detailed description of the evaluation based on the HGI benchmarks in the Methods section, line 676. We copied it below for your convenience.

“The evaluation was performed on the dataset HCD-EThcD-CID-MS/MS (file B from Kawahara et al.³, accession number PXD024101). N-linked and O-linked glycopeptide analyses were performed as follows. The protein database was obtained from the HGI study, which included 20,201 human proteins. The HGI study also provided two default glycan databases that contained 309 mammalian N-glycan compositions and 78 mammalian O-glycan compositions. Since GlycanFinder is a glycan structure based search engine, we collected all the glycan structures from our internal glycan database that had the same compositions as that of the HGI study. As a result, 494 N-glycan structures and 298 O-glycan structures in GlycoCT format were used to perform the HGI analysis with GlycanFinder. The protein and glycan databases are provided in the Supplementary Data 1. Other search parameters are as follows: EThcD fragmentation for N-glycan analysis and HCD fragmentation for O-glycan analysis; trypsin cleavage, semi-specific digestion with at most one missed cleavage; Cys (Carbamidomethylation, +57.02 Da) as fixed modification, Met (Oxidation, +15.99 Da) and Asn/Gln (Deamidation, +0.98 Da) as variable modifications; precursor error tolerance of 10 ppm, fragment error tolerance of 0.02 Da, glycan fragment error tolerance of 20 ppm.”

Reviewer #1 - comment 4

O-glycopeptide report Identification is missing in the Supplementary Table S5.

Authors' response:

We have added the O-linked results to the Supplementary Table S5, the table now includes both N-linked and O-linked results.

Reviewer #1 - comment 5

Supplementary Table S6 is just a copy of the information already provided in Kawahara et al and should be only referenced but not shown as supplementary.

Authors' response:

We have removed the Supplementary Table S6 as recommended.

Reviewer #1 - comment 6

Fig 6 is better placed as supplementary since the other analysis focused on N-glycopeptide only.

Authors' response:

We have moved Figure 6 to Supplementary Figure S7 as suggested,

Reviewer #1 - comment 7

MSFragger has been benchmarked by the HGI study and demonstrated better performance compared to what the authors showed in Figure 5 (<https://www.hupo.org/page-1757445>).

Likewise, the authors should consider benchmarking their software against the HGI study, a service they appear to offer, to avoid any bias or incorrect use of the score system. If the authors want to show a comparison against the HGI study it is recommended that only direct measurements are used (eg the authors could show a comparison of number of glycoPSMs reported by the HGI teams and PEAKS GlycanFinder, glycan composition correlation against a reference serum glycome, % of NeuGc and multi Fuc).

Authors' response:

We have been advised by Reviewer #2 below (Reviewer #2 - comment 1) that, according to the HGI study, each software should be run by its own developer team and no developer team should run another team's software to avoid conflict/bias. As a result, Reviewer #2 suggested us to remove the pGlyco3 and MSFragger results analyzed by our team (from Figure 5, Supplementary Figure S7, and Supplementary Table S5).

To avoid incorrect use of the score system in the HGI study, we have also reported direct measurements of GlycanFinder results in the Supplementary Table S5 as Reviewer #1 suggested. The glycoPSMs identified by GlycanFinder are also available in the Supplementary Table S5.

Reviewer #2 (Remarks to the Author):

Sun et al. have revised the manuscript describing their GlycanFinder software package, rerunning the comparison of the yeast FDR experiment and HGI benchmarking analysis and adding many more details about the method and IgG analysis. These revisions have greatly improved the paper and satisfied most of the issues I saw in the original manuscript.

Reviewer #2 - comment 1

I originally praised the authors for including the HGI benchmark for their software tool, and the addition of more detail on how the benchmark was performed is very welcome. However, the inclusion of the authors' attempts to run several other software tools and score them with the benchmark to show that GlycanFinder performed better is problematic and should be removed. In the HGI study, each software tool was run either by its own developers, or users of the software. In either case, the group submitting results was motivated to provide the best results possible, and a neutral third party assessed the performance. Notably, no developers ran another developer's software tool, because they would have a conflict/bias in wanting their software to perform the best (regardless of whether this was acted on). I do not mean to imply that the authors intended to bias the analysis, however, because this deviates considerably from how the actual benchmark was done, the other software tools analyzed by the GlycanFinder team should not be included in the benchmark figure.

Authors' response:

We have removed the pGlyco3 and MSFragger results analyzed by our team from Figure 5, Supplementary Figure S7, and Supplementary Table S5 as Reviewer #2 suggested. The main text has also been revised accordingly.

Reviewer #2 - comment 2

Additionally, insufficient detail is still provided to understand the HGI analysis even for GlycanFinder. What glycan database was used? The text states that all tools used the same database but not what that database was. This is crucial as using the final database obtained by the HGI study would be a significant (and potentially unfair) advantage for GlycanFinder over the other tools previously run in the benchmark.

Authors' response:

We have added the detailed description of the evaluation based on the HGI benchmarks in the Methods section, line 676. We copied it below for your convenience.

"The evaluation was performed on the dataset HCD-ET_hCD-CID-MS/MS (file B from Kawahara et al.³, accession number PXD024101). N-linked and O-linked glycopeptide analyses were performed as follows. The protein database was obtained from the HGI study, which included

20,201 human proteins. The HGI study also provided two default glycan databases that contained 309 mammalian N-glycan compositions and 78 mammalian O-glycan compositions. Since GlycanFinder is a glycan structure based search engine, we collected all the glycan structures from our internal glycan database that had the same compositions as that of the HGI study. As a result, 494 N-glycan structures and 298 O-glycan structures in GlycoCT format were used to perform the HGI analysis with GlycanFinder. The protein and glycan databases are provided in the Supplementary Data 1. Other search parameters are as follows: EThcD fragmentation for N-glycan analysis and HCD fragmentation for O-glycan analysis; trypsin cleavage, semi-specific digestion with at most one missed cleavage; Cys (Carbamidomethylation, +57.02 Da) as fixed modification, Met (Oxidation, +15.99 Da) and Asn/Gln (Deamidation, +0.98 Da) as variable modifications; precursor error tolerance of 10 ppm, fragment error tolerance of 0.02 Da, glycan fragment error tolerance of 20 ppm.”

We have also acknowledged in the Discussion section, line 410, that

“The evaluation benchmarks and results obtained from the HGI study by Kawahara et al.³ were published about a year before our study. Thus, this could be a disadvantage to those tools and new updates or progress might have been made during that period.”

Reviewer #2 - comment 3

The authors have provided additional details on how O-glycopeptides are analyzed, but some details remain unclear and need to be provided, and they have neglected to mention several key limitations.

- The authors state that GlycanFinder uses “internal” fragment ions to determine the location of O-glycans within the peptide, but it appears from supporting figure 6 that only the standard terminal ions are used (b/y and c/z). Internal ions are multiply fragmented sections of the peptide that contain neither terminus, resulting from a combination of b- and y-type fragmentation events (for example). These ions do exist in peptide fragmentation spectra at low levels, but they are unlikely to be of use in localization in most cases and have a significant potential for false matching given their much larger search space. I believe the authors did not mean to refer to internal ions, so please clarify if this is the case. If the authors do mean internal ions, how specifically are they being used (which ions are used? by and/or ay? cy, bz, cz? How often are they actually observed?) and what steps are being taken to account for the large search space and potential for false localization need to be addressed.

Authors’ response:

Unlike a peptide which is linear, a glycopeptide can be considered as two-dimensional with the peptide and the glycan dimensions. In this study, by “internal fragment ions” we refer to those ions that were fragmented more than once, such as a peptide ion with a glycan fragment attached, or a peptide internal ion with an intact glycan attached. In practice, GlycanFinder considers peptide ions with one monosaccharide attached (i.e. the glycan root) or with an intact

glycan attached when calculating the glycopeptide score and the A-score. The ion types depend on the fragmentation methods. For example, b/y ions + glycan fragments for HCD, b/y/c/z/z' ions + glycan fragments for EThCD, and c/z ions + glycan fragments for ETD fragmentation.

If the peptide of a glycoPSM has multiple possible glycosylation sites, a site-specific localization score, named A-score, is calculated by comparing those glycosylation sites and their respective internal fragment ions. In order to distinguish the ambiguity between potential glycosylation sites, fragment ions exclusive to a specific site need to be identified in order to assign a glycan to the residue. The A-score in GlycanFinder is calculated as $-10 \times \log_{10} P$, where the P value indicates the likelihood that the glycan site is assigned by chance. We adopted this idea from a previous study of phosphorylation site localization by Beausoleil et al.⁴⁶. More specifically, the probability of the correct glycosylation site is calculated based on the likelihood of identifying site-determining internal fragment ions compared to random chance. An A-score of 42.89 would represent a probability of less than 1 in 15,000 of matching a difference by random chance. If a glycan site can be inferred confidently, for instance, if there can be only one N-glycan site in a peptide, then the A-score would be 1000.

There are multiple serine and threonine residues in a protein that can be O-glycosylation sites. In order to keep the search space reasonable, GlycanFinder currently supports at most two O-glycans per peptide, it is also recommended to provide a glycoprotein database (less than 500 entries) instead of a large, entire proteome database. GlycanFinder considers internal fragment ions to determine the best glycosylation sites and calculates the site-specific localization score (A-score) to reflect the confidence of its glycosylation site assignments as described above (Supplementary Figure S6).

We have added the above details in a revised section “*Site-specific localization score (A-score) and O-linked glycopeptide analysis*” in the manuscript, line 469.

- No discussion of activation methods is provided, despite O-glycans generally requiring ETD (or similar) for localization. How are O-glycans localized in HCD scans by GlycanFinder? Figure S6 only shows EThcD scans. Or if they cannot be localized in HCD scans, what is reported for HCD scans searched for O-glycopeptides? How was this handled for the HGI benchmark?

Authors' response:

As mentioned above, different types of ions were considered depending on the fragmentation methods. For example, b/y ion + HexNAc fragments for HCD spectra, c/z ion + intact glycan for ETD spectra, and a mix of HCD and ETD fragment ions for EThcD spectra. If there are not enough ions that can be used to determine the best glycosylation site on a peptide, like in HCD spectra, it can still report a candidate based on peptide backbone ions and glycan B/Y ions. In this case, the A-score would be low, which indicates that the glycosylation site is not confident.

In the HGI study evaluation, the correctness of glycosylation sites is not considered in their scoring. Instead, they pay more attention to the accuracy of the identified peptide sequences,

the glycan compositions, as well as the identified glycoproteins. More specifically, the criteria of O-glycan tests in the HGI study include O-glycan composition (O1), Source O-glycoproteins (O2), Unique O-glycopeptides reported (O3), 'consensus' O-glycopeptides (O4), absence of NeuGc and multi-Fuc O-glycopeptides (O5).

- The ion types used for localization are not listed – are b/y ions used for localization? If so, what part of the glycan is being retained on these ions that can be used to localize (please show an example spectrum)? b/y+HexNAc fragments are quite rare for O-glycans (unlike N-glycans). If only c/z ions are used, please state that.

Authors' response:

As discussed above, the ion types used for localization depend on the fragmentation method used when generating the MS/MS spectra. Below are some example spectra:

HCD example: there are only a few of b/y ions with glycan fragments observed.

HGI file B, Scan 5629, A-score=0

ETHcD example: it has plenty of b/y/c/z/z' ions with glycan fragments or intact glycan attached. The glycosylation sites can be determined more confidently.

PXD020077, 2019_09_19_OPRmix_35trig_ETHcD35_rep2.raw, Scan 8552, A-score=17.02

Anal. Chem. 2020, 92, 22, 14878–14884

- The limitation of max 2 glycans per peptide is significant, as many O-glycopeptides contain more than 2 glycans. Many of the HGI benchmark study O-glycopeptides contained >2 glycans – how were these reported in the results, and how is GlycanFinder able to obtain such high scores despite not handling these glycopeptides?

Authors' response:

We agree with the reviewer that the option allowing >2 glycans per peptide may increase the chance of identifying more O-glycopeptides. However, it is not clear how much impact this option has on the final number of identifications, because different tools may also identify different peptides, not just O-glycans on the same peptides. According to the HGI study, the O3 test measures the number of unique O-glycopeptides reported by the search engines. We found that GlycanFinder reported 319 unique O-glycopeptides (Supplementary Table S5), while the best team in the HGI study (user team 13) reported 274 unique O-glycopeptides.

Reviewer #2 - comment 4

The comparison to StrucGP in Figure 3 is a welcome addition to the paper, and appears to show good performance for GlycanFinder as some de novo glycans are found that are missed by StrucGP. However, it is stated in the Discussion that the FDR for de novo glycans is not assessed. How were these discovered de novo glycans compared to StrucGP in a fair way? Either the FDR is not assessed (unlike StrucGP) and it is unclear if the additional glycans are correct or not, or the FDR is assessed in a second follow-up run. If the latter, this needs to be stated in the text so readers are not misled about how the analysis is performed and how it differs from the single run of StrucGP.

Authors' response:

In this revised manuscript, line 238, we have performed an FDR estimation of de novo glycans identified by GlycanFinder as follows:

“We also attempted to provide an FDR estimation for glycoPSMs identified from the glycan de novo sequencing. A proper way to do this is to add the de novo glycans to the original glycan database and then repeat the glycopeptide database search with the new glycan database and FDR control. In particular, we combined 922 de novo glycans identified by GlycanFinder from the testing data and 7,884 glycans in the original database, resulting in a new database of 8,806 glycans. A second round database search was then performed using the new database and 1% FDR. We found 1,948 additional glycoPSMs corresponding to 389 de novo glycans that passed 1% FDR of the second round database search, i.e. about 6.8% more glycoPSMs and 4.9% more glycans than the original database search results. Those extra de novo glycoPSMs are provided in the Supplementary Table S2.”

Reviewer #2 - comment 5

In both the abstract and introduction, the authors state that “Our results show that GlycanFinder not only controlled the FDRs accurately but also identified more glycopeptide-spectrum matches (glycoPSMs) than previously reported.” However, Fig 2 clearly shows GlycanFinder identifying fewer glycoPSMs than competing tools. Please adjust these statements to accurately reflect the data shown in the manuscript.

Authors' response:

We have revised the statements as follows:

“Our results show that GlycanFinder achieved comparable performance to other leading glycoproteomics softwares in terms of both FDR control and the number of identifications. Moreover, GlycanFinder was also able to identify new glycopeptides not found in existing databases. “

We have also added some additional results to the analysis in Figure 2 (main text, line 175). We copied it below for your convenience.

Since we noticed that our estimated FDRs were too tight, e.g. 0.1% for both glycan and peptide levels, we also attempted to relax the score thresholds so that our FDRs were comparable to those of MSFragger, i.e. 0.3% and 0.2% at glycan and peptide levels, respectively. With those comparable FDRs, GlycanFinder would identify 4,518 glycoPSMs, about 4.5% less than MSFragger. In addition, it is worth noting that MSFragger is based on glycan compositions, whereas GlycanFinder and pGlyco3 report glycan structures, which provide more comprehensive information about the glycans as one composition may correspond to multiple structures.

Reviewer #2 - comment 6

Page 4, line 70 – the authors state “GlycanFinder accurately estimates the FDRs at both peptide and glycan levels by ... designing proper decoys for those levels.” While the mass

shifted glycan FDR is the best method developed thus far, it is not a proper decoy as it significantly underestimates the rate of false matches to glycans with shared fragment ions (i.e., most glycans). Please note this general limitation of this glycan FDR method.

Authors' response:

We have added this limitation of this glycan FDR method in the manuscript, line 99.

Reviewer #2 - comment 7

Page 8, line 191 – which of the 5 mouse tissues was the one used for testing (vs the other 4 for training)? Are the results in Fig. 3 shown from the training or testing data (or both)?

Authors' response:

The results in Figure 3a were calculated from the testing data in a five-fold cross validation. For instance, when the lung data was used for testing, the other four (brain, kidney, heart, liver) were used for training. Similarly, when the brain data was used for testing, the other four (lung, kidney, heart, liver) were used for training. We have added this clarification in the revised manuscript, line 206.

Reviewer #3 (Remarks to the Author):

The manuscript describes GlycanFinder, a software program that can accurately identify and validate glycopeptides by combining database search and de novo sequencing techniques. When a glycopeptide cannot be determined from a database search, a deep learning approach is used to reconstruct glycan structures through de novo sequencing.

Reviewer #3 - comment 1

The performance of GlycanFinder is somewhat disappointing. Database search results in Figure 2 showed that MSFragger resulted in 17% more glycoPSMs when FDR was estimated at 1%. For the IgG dataset analysis, pGlyco3 identified 254 unique glycopeptides, while GlycanFinder identified 178.

Authors' response:

In this revision, we have added some additional results to the analysis in Figure 2 (main text, line 175). We copied it below for your convenience.

“Since we noticed that our estimated FDRs were too tight, e.g. 0.1% for both glycan and peptide levels, we also attempted to relax the score thresholds so that our FDRs were comparable to those of MSFragger, i.e. 0.3% and 0.2% at glycan and peptide levels, respectively. With those comparable FDRs, GlycanFinder would identify 4,518 glycoPSMs, about 4.5% less than MSFragger. In addition, it is worth noting that MSFragger is based on glycan compositions, whereas GlycanFinder and pGlyco3 report glycan structures, which provide more comprehensive information about the glycans as one composition may correspond to multiple structures.”

For the IgG dataset, pGlyco3 identified more glycopeptides than GlycanFinder (254 versus 178, Supplementary Figure S5). But for the fission yeast dataset in Figure 2, pGlyco3 had a higher glycan FDR (5.0%) and fewer glycoPSMs (3,553 versus 4,035) than GlycanFinder. We also noted that MSFragger only identified 88 glycopeptides from the IgG dataset, much fewer than GlycanFinder and pGlyco3. Thus, the results from the two datasets in Figure 2 and Supplementary Figure S5 show that the three glycopeptide search engines were competitive and there was no clear winner that outperformed the others.

Reviewer #3 - comment 2

In lines 197-203, the accuracy comparison between GlycanFinder and StrucGP was made and Supplementary Table S2 was referred to. However, the table provides identification results that cannot be directly compared.

Authors' response:

Supplementary Table S2 provides the glycoPSMs of the de novo results of GlycanFinder and StrucGP. The de novo accuracies in Figure 3a were calculated from the results in the Supplementary Table S2. A Python script and instructions are provided in our GitHub repository (<https://github.com/zqq66/GlycoNovo>) for the accuracy calculation.

Reviewer #3 - comment 3

De novo glycan sequencing results shown in Figure 3 gave better performance than StrucGP but its structural accuracy was just above 30%. Given that this performance was obtained through machine learning, it is difficult to believe that the training was successful.

In lines 219-220, it is stated that “the accuracy of glycan de novo sequencing was on par with that of de novo peptide sequencing”. What grounds did you use to make such a claim? Glycan de novo sequencing accuracy was slightly higher than 30%, significantly lower than the usual de novo peptide sequencing accuracy.

Authors' response:

When we proposed the first deep learning model for de novo peptide sequencing, DeepNovo (<https://doi.org/10.1073/pnas.1705691114>), it achieved 59% amino acid accuracy and 30% peptide accuracy on human peptides. Since then, the accuracy of de novo peptide sequencing has been steadily improved. A recent study by Rui et al. covering some popular tools for de novo peptide sequencing such as PEAKS, DeepNovo, PointNovo, SMSNet, pNovo, shows that the amino acid accuracy reached 70-80% and the peptide accuracy reached 40-60%, depending on the datasets (<https://doi.org/10.1038/s42256-021-00304-3>). In our study, the de novo glycan sequencing results show that GlycanFinder achieved about 83% fragment ion accuracy and 32% structural accuracy. However, it should be noted that the tree structure of glycans is much more complex than the linear structure of peptides. During the de novo sequencing of a glycan, there are many ways that a monosaccharide can be added to a partial tree, creating multiple branches, whereas for a peptide, amino acids are simply added to the sequence one after another. Thus, it is very hard to predict the entire structure of a glycan correctly. In fact, while GlycanFinder and StrucGP showed comparable accuracies of fragment ions and composition (>80%), the structure accuracy of GlycanFinder was substantially higher than that of StrucGP (32% versus 23%). Given these considerations, we think the performance of GlycanFinder on de novo glycan sequencing is satisfactory. To the best of our knowledge, our work is the first application of deep learning to de novo glycan sequencing, so we expect that more improvements with better accuracy will come soon.

However, we also agree with Reviewer #3 that the comparison statement “*the accuracy of glycan de novo sequencing was on par with that of de novo peptide sequencing*” is subjective and those two problems are not directly comparable. So we have removed that statement. Instead, we added the following sentences to provide readers with some context (line 248):

“A recent study by Rui et al.²⁶ shows that, for de novo peptide sequencing, the amino acid accuracy could reach 70-80% and the peptide accuracy could reach 40-60%. However, it should be noted that the tree structures of glycans are much more complex than the linear structure of peptides. During the de novo sequencing of a glycan, there are many ways that a monosaccharide can be added to a partial tree, creating multiple branches, whereas for a peptide, amino acids are simply added to the sequence one after another.”

Reviewer #3 - comment 4

However, as I previously stated in my review comment, the description of the training model is only at a very abstract level, with insufficient details. It must include (1) model architecture (how many layers, how many units, any residual blocks, etc.) (2) the input representation and its dimension (3) hyperparameters (4) the final size of learnable parameters(weights) (5) training vs. validation errors during training, at the very least.

Authors' response:

We have revised the section “Deep learning model to construct the glycan tree” (line 518) to provide the details of our model training for de novo glycan sequencing. We copied it below for your convenience:

“Deep learning model to construct the glycan tree

Once the composition is determined, the glycan tree is constructed from the asparagine (Asn) residue of the peptide, i.e. the root, to leaves by adding various monosaccharides iteratively. At each iteration, a combination of monosaccharides is selected by a deep learning model and attached to a leaf node of the sub-tree obtained from the previous iteration. It should be noted that Figure 1c only shows a simple example where only one monosaccharide is attached to a leaf node. When a combination consisting of two or more monosaccharides is attached to a leaf node, it will create a branch at that node. The iterative process of adding such combinations will result in a branched topology of the glycan tree where multiple monosaccharides can be linked to any nodes in the tree. The set of all possible combinations of monosaccharides are derived from the training data. The training data in this study consists of twenty-one combinations of five monosaccharides (Hex, HexNAc, Fuc, NeuAc, NeuGc), along with a unique token indicating that no additional monosaccharide could be appended to the leaf node.

The deep learning model consists of two neural networks. The first neural network is applied to encode topologic knowledge of the sub-tree. We use the Graphormer architecture¹⁷ which inherits from the Transformer¹⁶ and is designed for graphs. The glycan structure is represented as a graph with nodes as monosaccharides and edges as linkages, and each of these nodes and edges is assigned a vector embedding. We encode each node in breadth-first search order with derived compositions from dynamic programming without the accumulated monosaccharides. The representations are then passed through two layers and four attention heads to enable the model to learn the relationships between different nodes and edges in the graph. The positional-embedding layer is also included to allow the model to learn the relative

positions of the nodes and edges. The output of this component is a 512-dimensional vector as graph embeddings, which captures the encoded glycan structure.

The second neural network captures the matched glycopeptide Y ions between the candidate trees and the spectrum. The theoretical spectrum is formed by attaching all candidates to the previously predicted substructure at each time step. The observed spectrum is represented by a set of m/z values and intensity pairs. Similar to PointNovo26, the pairs are encoded with respect to the mass gap between theoretical m/z and observed m/z . The differences between observed peaks and theoretical peaks are obtained as input features to the point-cloud model named T-Net. The permutation of its monosaccharides is used for each candidate combination as features. For example, the last child monosaccharides of the bisect core structure (Hex2HexNAc1) will have features of Hex, HexNAc, HexHexNAc, HexHex, Hex2HexNAc. At each prediction step, we obtain the theoretical m/z values for candidates and their permutations to compute the m/z difference tensor (D) for the observed peaks. We adopt the activation function from PointNovo to extract features from the spectrum:

$$\sigma(D) = \exp[-|D| \cdot c]$$

The spectrum features are also encoded into 512-dimensional vectors to ensure a fair concatenation of results from the two neural networks.

Finally, representations learned by the two neural networks are concatenated together and fed into a fully-connected layer which adjusts the weights between the two networks. The total number of parameters is 10,939,134. During training, a batch size of 256 and 10 epochs are used with 8-1-1 train-valid-test ratio. The best model is selected based on the least validation error. The model was trained and tested on 25 fractions, with each tissue (mouse brain, lung, liver, kidney, and heart) containing five fractions, in a five-fold cross-validation fashion: the data of four tissues were used for training and the data of the remaining tissue were used for testing. The training loss decreased from 0.220 to 0.105, while the validation loss also decreased to 0.251. During prediction (de novo sequencing), the two neural networks are used to sort candidates at each time step and the candidate that has the highest rank and contains only the monosaccharides found in the composition (predicted by dynamic programming approach) is selected. The candidate is then attached to the sub-tree and fed back to the neural networks for the next iteration. The iterations continue until the constructed tree includes all monosaccharides in the composition.

Data and model training

To train the neural networks, we ran GlycanFinder database search on the dataset of five mouse tissues (brain, heart, kidney, liver, lung) and identified 139,208 glycoPSMs at 1% FDR. However, we found that the number of unique glycan structures was limited, e.g. the mouse brain tissue only contained 1,155 unique glycan structures. Learning topologic knowledge on a thousand structures is not sufficient to generalize to other tissues. As a result, we divided the training process into two phases. The first phase is to take advantage of the glycan database and select 5012 unique structures to train the first deep learning model. To guarantee fair comparison, we excluded all glycan structures that exist in the testing set from the training set.

For each glycan structure, we iteratively removed all leaves attached to its parent monosaccharides from the first parent node to the root (i.e. in the opposite direction to that of the de novo sequencing). At each iteration, the partition created a subtree and the candidate (i.e. a set of monosaccharides) that had been removed. The second phase used the trained parameters from the first deep learning model and trained the second neural network on the spectrum to obtain the next monosaccharides group from 21 candidates. The predicted and the target monosaccharides were used to calculate the training loss, which was then back-propagated to update the parameter weights of the two neural networks.”

Reviewer #3 - comment 5

While the authors emphasized the importance of accurate FDR estimation, they did not describe how it is done, omitting how peptide and glycan scores are calculated. In the first place, it must be precisely defined, and the score distributions between target and decoy PSMs must be presented.

Authors' response:

We have provided more details of our calculation of peptide and glycan scores in the revised manuscript, line 430. We have also provided in the Supplementary Figure S1a an example of the score distributions of target and decoy glycoPSMs for FDR calculation on the mouse brain dataset (PXD005411) from Liu et al.²⁷. We copied them below for your convenience.

“Peptide-based and glycan-based searches are performed to generate candidate glycopeptides. During the peptide-based search, possible glycan peaks are first removed from a glycopeptide spectrum to avoid the influence on the peptide score calculation. Fragment ions at the lower end of a spectrum are quickly searched against peptide backbone ions, including b/y, b/y + HexNAc for HCD and c/z for ETD data, to identify candidate peptides. We also consider the ions with losses of ammonia (-NH₃) or water (-H₂O), and charge (2+). Each spectrum is matched with the in silico digested peptides according to a peptide scoring function, with glycans treated as PTMs. The peptide scoring function is similar to the normal PEAKS DB search scoring. It uses a linear discriminative function (LDF) score to measure the quality of a peptide-spectrum match. After a peptide with a glycan mass offset is selected based on the peptide score, glycan candidates can be obtained from the glycan database according to the mass. Glycopeptide Y ions and B ions are then used to calculate the glycan score. The glycan scoring function is obtained by training by an XGBoost regression model on a variety of published datasets. Tens of features are tuned to evaluate a glycan-spectrum match. Several key features include “ratio of observed glycan Y-ions”, “log₁₀ Y-ions intensity”, “observed core structure”, “ratio of observed glycan B-ions” and etc.

If the spectrum cannot be identified by the peptide-based search, it proceeds to the glycan-based search. Here the fragment ions at the higher end of the spectrum are quickly searched against glycopeptide Y ions to identify candidate glycans. Subsequently, candidate peptides are deduced from the precursor mass, candidate glycans, and the protein database.

The peptide mass is further added to glycan fragments when matching glycan Y ions with the spectrum. Glycopeptide-spectrum matches are then evaluated using the same peptide and glycan scoring functions as in the peptide-based search.”

Supplementary Figure S1a below shows an example of the score distributions of target and decoy glycoPSMs for FDR calculation on the mouse brain dataset (PXD005411) from Liu et al.²⁷.

Reviewer #3 - comment 6

In the same vein, it is not described how A-score is computed by GlycanFinder. It is vaguely described as “comparing glycosylation sites and their respective internal fragment ions”. By the way, how do you define “internal fragment ion” in a glycoPSM?

Authors' response:

We have provided more details of the A-score calculation in the revised manuscript, line 469. We copied it here for your convenience.

“If the peptide of a glycoPSM has multiple possible glycosylation sites, a site-specific localization score, named A-score, is calculated by comparing those glycosylation sites and their respective internal fragment ions. In order to distinguish the ambiguity between potential glycosylation sites, fragment ions exclusive to a specific site need to be identified in order to assign a glycan to the residue. In this study, by “internal fragment ions” we refer to those ions that were fragmented more than once, such as a peptide ion with a glycan fragment attached, or a peptide internal ion with an intact glycan attached. In practice, GlycanFinder considers peptide ions with one monosaccharide attached (i.e. the glycan root) or with an intact glycan attached

when calculating the glycopeptide score and the A-score. The ion types depend on the fragmentation methods. For example, b/y ions + glycan fragments for HCD, b/y/c/z/z' ions + glycan fragments for EThCD, and c/z ions + glycan fragments for ETD fragmentation.

The A-score in GlycanFinder is calculated as $-10 \times \log_{10} P$, where the P value indicates the likelihood that the glycan site is assigned by chance. We adopted this idea from a previous study of phosphorylation site localization by Beausoleil et al. (Beausoleil et al. 2006). More specifically, the probability of the correct glycosylation site is calculated based on the likelihood of identifying site-determining internal fragment ions compared to random chance. An A-score of 42.89 would represent a probability of less than 1 in 15,000 of matching a difference by random chance. If a glycan site can be inferred confidently, for instance, if there can be only one N-glycan site in a peptide, then the A-score would be 1000."

Reviewer #3 - comment 7

Contrary to the original purpose of A-score for PTM site localization, it appears that A-score is simply included as an output of GlycanFinder and is not used to confidently localize glycosylation sites. Why?

Authors' response:

GlycanFinder uses peptide and glycan FDRs to control the quality of identification results. Glycan localization is not the most important factor when calculating the score of a glycopeptide, since it can be ambiguous. A-score provides a measurement of the site confidence, which can be utilized by users to further filter the results based on a specified threshold.

Reviewer #3 - comment 8

When estimating an FDR for IgG data, the database consisted of 9 protein sequences and 247 glycans. What statistical significance do you believe a target-decoy validation has when such a tiny database is used?

Authors' response:

GlycanFinder estimates the FDRs at both peptide and glycan levels. For peptide FDR, a decoy protein database is generated by randomly shuffling the target protein database. In order to validate the FDR accuracy on the limited size of this IgG protein database, we have conducted an additional experiment on the IgG Orbitrap dataset. In this experiment, the decoy protein database was enlarged to 25 times the size of the target IgG protein database and contained nearly 50,000 amino acids. Using this new decoy protein database, we found that the number of glycoPSMs identified at 1% FDR only dropped slightly from 2,215 to 2,192, indicating that the GlycanFinder results are still reliable even with a small protein database. For glycan FDR, we apply random mass shifts of 3-30 m/z to all fragment ions in an MS2 spectrum to create a decoy spectrum. The number of decoys in the glycan FDR calculation is the number of glycoPSMs

with decoy spectra, thus we believe it is correlated with the number of spectra and it is not affected by the size of the glycan database.

Reviewer #3 - comment 9

It is not clear how a glycan is sequenced de novo. According to Figure 1c, a monosaccharide appears to be selected and it can only be attached to a leaf node, not an internal node of a tree. If that's the case, I am not sure if we can call it a "true" de novo glycan sequencing. If not, it must be clearly stated to what extent novel glycan structures can be deduced from tandem mass spectra.

Authors' response:

We have added the following details in the revised manuscript, line 518, to explain how a combination of monosaccharides is added at each iteration and how the iterative sequencing process will create branches and different structures of a glycan tree.

"... the glycan tree is constructed from the asparagine (Asn) residue of the peptide, i.e. the root, to leaves by adding various monosaccharides iteratively. At each iteration, a combination of monosaccharides is selected by a deep learning model and attached to a leaf node of the sub-tree obtained from the previous iteration. It should be noted that Figure 1c only shows a simple example where only one monosaccharide is attached to a leaf node. When a combination consisting of two or more monosaccharides is attached to a leaf node, it will create a branch at that node. The iterative process of adding such combinations will result in a branched topology of the glycan tree where multiple monosaccharides can be linked to any nodes in the tree. The set of all possible combinations of monosaccharides are derived from the training data. The training data in this study consists of twenty-one combinations of five monosaccharides (Hex, HexNAc, Fuc, NeuAc, NeuGc), along with a unique token indicating that no additional monosaccharides could be appended to the leaf node."

Reviewer #3 - comment 10

In Figure 4, the authors emphasized the challenge of distinguishing the four different isomeric glycopeptides of IgG1-4. It is not clear why this is a challenge. The four peptides from IgG1 vs. IgG2 vs. IgG3/IgG4 have different parent masses. Any algorithm could distinguish between IgG3 and IgG4 if their backbone fragmentations were successful, as shown in Fig. 4b.

Authors' response:

Human IgG all carry a conserved N-glycosylation site in the Fc domain and this site is occupied by an N-linked glycan in more than 99% of the IgG molecules. Due to the high similarity of the sequences among all 4 IgG subclasses, it is hard to separate by LC. Previous methods usually conducted antibody-based pre-separation, chemical labeling or different fragmentation methods

to distinguish them. Besides, the mass difference between Phe and Tyr residues has exactly the same mass difference as swapping monosaccharides Hex and Fuc. GlycanFinder provides comprehensive glycosylation analysis including both the peptide and glycan identification. In particular, the glycopeptide B/Y ions with higher masses and intensities allowed to identify the glycan structure (revised Supplementary Figure S3a), whereas the peptide backbone b/y ions with lower masses and intensities helped to identify the peptide sequences (revised Figure 4a, Supplementary Figure S3b).

Reviewer #3 - comment 11

Figure 4b: for both mass spectra of peptides “EEQYNSTFR” and “EEQFNSTYR” at the bottom panel, two fragment ions flanking the annotated amino acid in blue letters (F and Y, respectively) are not clearly shown.

Authors' response:

We have doubled the sizes of the annotated spectra in Figure 4 so that they are better displayed. We have also provided the Supplementary Figure S3b showing the details of the matched fragment ions of each amino acid. For instance, the letter F of IgG3 peptide is supported by two blue ions, b8 (mass 999.41 Da) and b8-NH3 (mass 982.39 Da), and four red ions y1 (mass 175.12 Da), y1-NH3 (mass 158.09 Da), y2 (mass 322.19 Da) and y2-NH3 (mass 305.16 Da)

Reviewer #4 (Remarks to the Author):

Reviewer #4 - comment 1

Although some of the comments were addressed, the authors were not convincing in conveying that the current developments make a significant change to existing methods in terms of user friendliness, adaptability and performance, additionally the representation of the work is not sufficient.

For example, what are the improvements when compared to recently published tools in Nature Communications, such as pGlycoQuant (<https://www.nature.com/articles/s41467-022-35172-x>)? This tool offers solutions to most of the described issues, and more, including MS1-based identification and quantification. The comparison between the current tool and pGlyco3 and MSFragger shows similar performance, it is unclear what is the advantage of the current tool.

Authors' response:

We thank Reviewer #4 for your comments. In this study, we have performed extensive validation on multiple datasets, including the fission yeast, the five mouse tissues, the IgG, and the HGI data. The results demonstrate that GlycanFinder shows competitive performance to other leading glycoproteomics search engines such as pGlyco3 or MSFragger. Another major contribution of GlycanFinder is the de novo glycan sequencing, which is not available in many other glycoproteomics softwares. We also show that GlycanFinder could predict the structures of de novo glycans more accurately than a recent method StrucGP. Thus, we believe the advantages offered by GlycanFinder will be beneficial to the glycoproteomics research community.

We also thank Reviewer #4 for the information about the recent publication of pGlycoQuant. We have acknowledged in the Discussion (line 400) that

“Our study did not include a comprehensive evaluation of glycopeptide quantitative analysis. In addition to the identification, accurate quantification of intact glycopeptides is essential for differential analysis of site-specific glycosylation. For instance, a recent deep learning-based method, pGlycoQuant⁴⁵, has been proposed for intact glycopeptide quantitation and can be used with common search engines such as pGlyco3, MSFragger, or Byonic for quantitative glycoproteomics.”

Reviewer #4 - comment 2

Still, the figure quality is low: There is a lot of unused white spacing, alterations in font size and color. Most graphs are directly copied from excel, without careful formatting, some mass spectra are so small that they are unreadable. The figures do not meet the high-quality standard of figures for publication in the Nature journals.

Authors' response:

In this revised manuscript, we have improved the quality of the figures as the Reviewers suggested. In particular, we have doubled the sizes of the annotated spectra and glycans so that they can be clearly displayed. The spectra are also annotated with their scan numbers, the glycan compositions are also displayed next to the glycan cartoons. We have carefully formatted the charts, the font size, the x- and y-axis, the legends. Charts are supplied in vector format wherever it is possible, other charts are supplied with at least 300 dpi as required by the journal. We have also tried our best to reorganize the figures to reduce the white space.

Reviewer #4 - comment 3

It is still unclear how quantification is performed. What do the authors mean by “the area of the isotope feature detected in the LC-MS scan and associated with that MS/MS spectrum is used for quantification of that glycoPSM. The quantification of each glycopeptide is calculated from all of its glycoPSMs”. I do not expect glycoPSMs to be quantified individually, rather used for the identification of glycopeptides. The relevant MS1 traces of the glycopeptide (including relevant isotopes and charge states) should then be used for quantification. Please describe how this is approached, or if a completely different approach is taken.

Authors' response:

We have added a section named “GlycanFinder quantification of glycopeptides” in the Methods section, line 588. We copied it below for your convenience. We believe that our quantification approach is similar to what you mentioned in your comments.

“GlycanFinder quantification of glycopeptides

For each identified glycopeptide, the sum of the areas of its MS1 precursor features with different charges is used for its quantification. The label-free approach is applied for glycopeptide quantification among different samples. Given a glycosylation site of a glycoprotein, the glycan profile is calculated based on the normalized areas of the glycan forms at that site. First, the area of each glycan form at each glycosylation site is calculated by summing the areas of all glycopeptides containing that glycan at that site. Then, the normalized area of each glycan form is obtained by dividing its area by the sum of the areas of all glycans forms at that site.”

Reviewer #4 - comment 4

It is important to discuss memory and run time in the main text and compare this to other tools.

Authors' response:

We have performed an additional analysis to compare the running times of GlycanFinder, pGlyco3, MSFragger, and MetaMorpheus on the fission yeast dataset (PXD005565). The

analysis was performed on a Dell Precision 7920 Tower with Intel Xeon and 128G RAM. All four tools support multiprocessors to accelerate the search speed and 36 processors were used in this analysis. All tools read in the same raw files and searched the same protein database and glycan database. Their running times are reported in Supplementary Table S1 (copied below) and discussed in the revised manuscript, line 194. The results show that MSFragger was the fastest tool and took 30 minutes to finish the analysis. GlycanFinder and pGlyco3 completed the analysis in 78 and 148 minutes, respectively, while MetaMorpheus took much longer time.

Software	Runtime (min)
GlycanFinder	78.18
MSFragger	30.0
pGlyco3	148.1
Meta Morpheus	689.13

Reviewer #4 - comment 5

It is unclear how “This approach guarantees that we do not miss any candidates, as opposed to using one strategy alone”. How can you prove that no candidates are missed?

Authors’ response:

We agree with the Reviewer that we cannot prove that no candidates are missed. We have revised the statement in line 58 as

“This approach reduces the chance that some candidates may be missed due to poor signals of peptide or glycan fragment ions, as opposed to using one strategy alone”.

Reviewer #4 - comment 6

It is unclear how glycan and peptide scores are determined, what calculations are made to come to these scores?

Authors’ response:

We have provided more details of our calculation of peptide and glycan scores in the revised manuscript, line 430. We have also provided in the Supplementary Figure S1a an example of the score distributions of target and decoy glycoPSMs for FDR calculation on the mouse brain dataset (PX005411) from Liu et al.²⁷. We copied them below for your convenience.

“Peptide-based and glycan-based searches are performed to generate candidate glycopeptides. During the peptide-based search, possible glycan peaks are first removed from a glycopeptide spectrum to avoid the influence on the peptide score calculation. Fragment ions at the lower end of a spectrum are quickly searched against peptide backbone ions, including b/y, b/y + HexNAc for HCD and c/z for ETD data, to identify candidate peptides. We also consider the ions with

losses of ammonia (-NH₃) or water (-H₂O), and charge (2+). Each spectrum is matched with the *in silico* digested peptides according to a peptide scoring function, with glycans treated as PTMs. The peptide scoring function is similar to the normal PEAKS DB search scoring. It uses a linear discriminative function (LDF) score to measure the quality of a peptide-spectrum match. After a peptide with a glycan mass offset is selected based on the peptide score, glycan candidates can be obtained from the glycan database according to the mass. Glycopeptide Y ions and B ions are then used to calculate the glycan score. The glycan scoring function is obtained by training by an XGBoost regression model on a variety of published datasets. Tens of features are tuned to evaluate a glycan-spectrum match. Several key features include “ratio of observed glycan Y-ions”, “log₁₀ Y-ions intensity”, “observed core structure”, “ratio of observed glycan B-ions” and etc.

If the spectrum cannot be identified by the peptide-based search, it proceeds to the glycan-based search. Here the fragment ions at the higher end of the spectrum are quickly searched against glycopeptide Y ions to identify candidate glycans. Subsequently, candidate peptides are deduced from the precursor mass, candidate glycans, and the protein database. The peptide mass is further added to glycan fragments when matching glycan Y ions with the spectrum. Glycopeptide-spectrum matches are then evaluated using the same peptide and glycan scoring functions as in the peptide-based search.”

Supplementary Figure S1a below shows an example of the score distributions of target and decoy glycoPSMs for FDR calculation on the mouse brain dataset (PXD005411) from Liu et al.²⁷.

REVIEWERS' COMMENTS

Reviewer #1 (Remarks to the Author):

The authors put significant effort to improve the manuscript and address all the reviewer's comment. Site localization accuracy for O-glycopeptides was not comprehensively assessed in this study. I would suggest the authors performing a follow-up study designed to assess the site localization score feature of GlycanFinder, which can be very helpful for the community. O-glycopeptide libraries are commercially available and could be used as good standards for this assessment. This manuscript is suitable for publication.

Reviewer #2 (Remarks to the Author):

The authors have made a second round of extensive changes that have now resolved the issues noted in the original and first-revision versions of the manuscript. I am satisfied that the manuscript is now suitable for publication.

Reviewer #3 (Remarks to the Author):

1. I am not sure if the method can be called "de novo", when it considers only 21 combinations of five monosaccharides. What are those 21 combinations?

2. It would be better to include a figure depicting the machine learning model with its inputs and outputs as well as its architecture.

3. The FDR estimation used in the manuscript is problematic in many ways:

3-1. Decoy generation method adopted in this manuscript does not seem right. A decoy spectrum generated by random-mass-shift must be shown that it matches both target & decoy glycopeptides with equal chances. Supplementary Figure 1 shows that random matches are not equally distributed between target and decoy.

3-2. In the rebuttal, it is claimed that, when compared with MSFragger, even if GlycanFinder gives a smaller number of glycoPSMs than MSFragger, it is superior to MSFragger considering the fact that it gives glycan structures as well as its compositions. But the manuscript did not describe the verification method for glycan structures returned by GlycanFinder, except for the fission yeast dataset where the verification was based on the composition (i.e., rejecting those with monosaccharides not expected in fission yeast). How does your Glycan score distinguish different glycan structures of the same glycan composition? Without knowing that glycoPSMs after FDR estimation would be correct in terms of their glycan structure, it cannot be considered superior to MSFragger. Returning a structure is one thing and its validation is a different story.

3-3. In proteomics community, a database of 50000 amino acids (probably about 100 protein sequences) is simply not acceptable for a target-decoy method. It cannot generate a sufficient number (and variety) of random matches that forms the basis of FDR estimation.

Point-by-point Response Letter

We thank the Reviewers for your constructive comments that greatly helped us to improve our manuscript. We have revised our manuscript and fully addressed all of your concerns. For more details, please see our point-by-point responses to all of your questions below. Our responses and all revision changes in the manuscript are marked in blue color for your convenience.

Reviewer #3 (Remarks to the Author):

The manuscript describes GlycanFinder, a software program that can accurately identify and validate glycopeptides by combining database search and de novo sequencing techniques. When a glycopeptide cannot be determined from a database search, a deep learning approach is used to reconstruct glycan structures through de novo sequencing.

Reviewer #3 - comment 1

I am not sure if the method can be called “de novo”, when it considers only 21 combinations of five monosaccharides. What are those 21 combinations?

Authors' response:

As we have mentioned in the Methods section, the set of all possible combinations of monosaccharides are derived from the training data. The training data in this study consists of twenty-one combinations of five monosaccharides (Hex, HexNAc, Fuc, NeuAc, NeuGc). The monosaccharides and their combinations can be updated when the training data is updated. They are not limited to just those presented in this study.

Reviewer #3 - comment 2

It would be better to include a figure depicting the machine learning model with its inputs and outputs as well as its architecture.

Authors' response:

We have added the Supplementary Figure S8 to depict our deep learning model for glycan de novo sequencing. Below is a copy for your convenience.

In a manner similar to RNN settings, our approach involves the extraction of information from both the spectrum and the current substructure at each time step. Each node within the substructure represents compositions derived from dynamic programming. For instance, <sos> in the initial time step consists of node features of two Hexes and three HexNAcs, the HexNAc node in the second time step consists of two Hexes and two HexNAcs. These combinations of monosaccharides are integrated into the substructure. The generation process concludes either when all monosaccharides in composition have been utilized or when all child nodes are connected by a <stop> sign.

We took advantage of the state of the art, Graphormer, to encode structure at each time step. Each node in structure is labeled by breadth-first-search order starting from the root attached to the peptide. As a result we use positional embedding introduced in the transformer to encode the label and centrality encoding that encodes the in and out degrees of each node.

Reviewer #3 - comment 3

The FDR estimation used in the manuscript is problematic in many ways:

3-1. Decoy generation method adopted in this manuscript does not seem right. A decoy spectrum generated by random-mass-shift must be shown that it matches both target & decoy glycopeptides with equal chances. Supplementary Figure 1 shows that random matches are not equally distributed between target and decoy.

Authors' response:

We have acknowledged the limitations of the decoy generation method in line 101: “ It should be noted that while this random-mass-shift approach is commonly used, it may underestimate the rate of false matches to glycans with shared fragment ions. Better approaches to generate decoys for glycan FDR estimation will be an interesting topic for future research.”

3-2. In the rebuttal, it is claimed that, when compared with MSFragger, even if GlycanFinder gives a smaller number of glycoPSMs than MSFragger, it is superior to MSFragger considering the fact that it gives glycan structures as well as its compositions. But the manuscript did not

describe the verification method for glycan structures returned by GlycanFinder, except for the fission yeast dataset where the verification was based on the composition (i.e., rejecting those with monosaccharides not expected in fission yeast). How does your Glycan score distinguish different glycan structures of the same glycan composition? Without knowing that glycoPSMs after FDR estimation would be correct in terms of their glycan structure, it cannot be considered superior to MSFragger. Returning a structure is one thing and its validation is a different story.

Authors' response:

Certain fragment ions in glycopeptide mass spectra do provide some information about the glycan structures (although such information is not always complete). Hence, GlycanFinder is designed to search glycan databases with structure information (e.g. in GlycoCT format) instead of using only glycan compositions. When doing so, GlycanFinder can consider some fragment ions that can help to distinguish one structure from another (with the same composition), as illustrated in the example below. When such fragments ions are available, they are included in the calculation of the glycan score and the glycan FDR estimation. When the fragment ions are not enough to completely resolve different structures with the same composition, GlycanFinder provides an S-score, which is calculated as the score difference between the best and second-best glycan structures. The purpose of this is to inform that (1) there are different glycan structures with the same composition that can match the given spectrum; (2) the fragment ions are not enough to completely resolve the structures; and (3) the S-score implies how much one structure is better than the other. Thus, this is a feature of GlycanFinder to provide more information about possible glycan structures for a given spectrum, as we noted in the manuscript, line 180. We do not claim that this makes GlycanFinder more accurate or superior to MSFragger, and we do not have validation that a structure returned with a high S-score is always 100% correct.

3-3. In proteomics community, a database of 50000 amino acids (probably about 100 protein sequences) is simply not acceptable for a target-decoy method. It cannot generate a sufficient number (and variety) of random matches that forms the basis of FDR estimation.

Authors' response:

We agree with Reviewer #3 that, for general proteomics, a large protein database is needed to achieve the statistical power for target-decoy differentiation. However, for some special applications such as antibody characterization, the databases are often very limited. Most available antibody databases only have a couple of hundreds sequences, or even less. From our experience, in such applications, using a small database with a few hundreds protein sequences still provides a decent FDR estimation with the target-decoy method. We shall continue to improve our method and explore other possible approaches to better address such situations.